# A Commander-independent function of COMMD3 in endosomal trafficking

**Galen T Squiers, Chun Wan, James Gorder, Harrison Puscher[†], Jingshi Shen***

Department of Molecular, Cellular and Developmental Biology, University of Colorado, Boulder, United States

## eLife Assessment

This **important** study explores the mechanisms underlying the maintenance of cell surface protein levels. The authors present **solid** evidence to support their claims, though the addition of certain validation experiments could have further strengthened the conclusions. This work will be of particular interest to cell biologists focused on membrane trafficking.

**\*For correspondence:**
jingshi.shen@colorado.edu

**Present address:** [†]Department of Biological Sciences, University of Southern California, Los Angeles, United States

**Competing interest:** The authors declare that no competing interests exist.

**Abstract** Endosomal recycling is a branch of intracellular membrane trafficking that retrieves endocytosed cargo proteins from early and late endosomes to prevent their degradation in lysosomes. A key player in endosomal recycling is the Commander complex, a 16-subunit protein assembly that cooperates with other endosomal factors to recruit cargo proteins and facilitate the formation of tubulo-vesicular carriers. While the crucial role of Commander in endosomal recycling is well established, its molecular mechanism remains poorly understood. Here, we genetically dissected the Commander complex using unbiased genetic screens and comparative targeted mutations. Unexpectedly, our findings revealed a Commander-independent function for COMMD3, a subunit of the Commander complex, in endosomal recycling. COMMD3 regulates a subset of cargo proteins independently of the other Commander subunits. The Commander-independent function of COMMD3 is mediated by its N-terminal domain (NTD), which binds and stabilizes ADP-ribosylation factor 1 (ARF1), a small GTPase regulating endosomal recycling. Mutations disrupting the COMMD3-ARF1 interaction diminish ARF1 expression and impair COMMD3-dependent cargo recycling. These data provide direct evidence that Commander subunits can function outside the holo-complex and raise the intriguing possibility that components of other membrane trafficking complexes may also possess functions beyond their respective complexes.

## Introduction

Proteins embedded in the plasma membrane transduce signals, import nutrients, and sense physical changes, enabling cells to respond to external cues (*Bonifacino and Glick, 2004*; *Rothman, 2014*; *Schekman and Novick, 2004*). The cell has evolved complex and interconnected mechanisms to maintain membrane protein levels on the cell surface and to adjust these levels in response to stimuli (*Bonifacino and Glick, 2004*; *Bausch-Fluck et al., 2018*). Disruption of membrane protein homeostasis underlies many human diseases, including cancer, metabolic disorders, and neurodegeneration (*Dell'Angelica and Bonifacino, 2019*; *Healy et al., 2023*; *Boesch et al., 2024*). Surface levels of a membrane protein are established through exocytosis, a vesicle fusion event, and internalization via endocytosis, a vesicle budding process (*Bonifacino and Glick, 2004*; *Wang et al., 2023*; *Gulbranson et al., 2019*). In addition to exocytosis and endocytosis, another branch of intracellular membrane trafficking – endosomal recycling – also plays a crucial role in maintaining surface protein homeostasis (*Simonetti and Cullen, 2019*; *Cullen and Steinberg, 2018*; *Yong et al., 2023*; *Wang et al., 2018*;

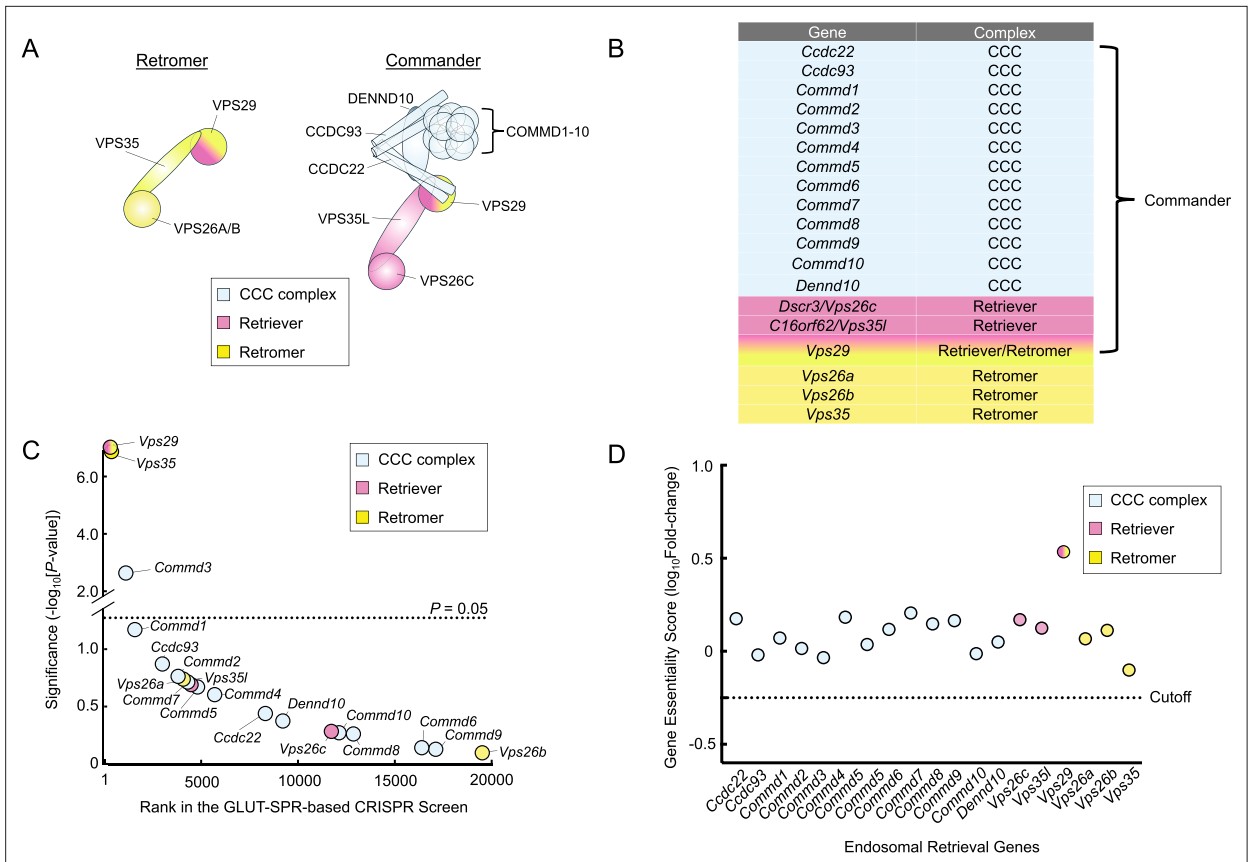

**Figure 1.** Genetic analysis of genes encoding Retromer and Commander subunits using unbiased genome-wide CRISPR screens. (**A**) Cartoon representation of the Retromer and Commander complexes. (**B**) Table listing genes encoding the subunits of the Retromer and Commander complexes. (**C**) Ranking of genes encoding the Retromer and Commander complexes in an unbiased CRISPR screen that was conducted to identify genes required for the surface homeostasis of GLUT-SPR (***Wang et al., 2023***). The GLUT-SPR reporter was constructed by inserting a hemagglutinin (HA) epitope into an exoplasmic loop of the glucose transporter GLUT4, with a GFP tag fused to the intracellular C-terminus of GLUT4 (***Gulbranson et al., 2017***; ***Klip et al., 2019***; ***Blot and McGraw, 2008***). Surface expression (HA staining) of the reporter was normalized to total reporter expression (GFP fluorescence) as a measure of relative surface levels of the reporter. Each dot represents a gene. The dashed line depicts the $P$-value cutoff at 0.05. (**D**) Essentiality scores of genes encoding the Retromer and Commander complexes were calculated by comparing gRNA abundance in a passage cell population (without any selection) with that in the initial CRISPR library. Genes with essentiality scores below the horizontal cutoff line are predicted to be essential to cell survival or growth. Full datasets of the CRISPR screens are included in a previous report (***Wang et al., 2023***).

The online version of this article includes the following source data for figure 1:

**Source data 1.** CRISPR screen data ranked by significance is shown in **Figure 1C**.

**Source data 2.** Gene essentiality scores are shown in **Figure 1D**.

***Ambrosio et al., 2022***). Following endocytosis, endocytic vesicles fuse into early endosomes, which then progress into late endosomes (***Cullen and Steinberg, 2018***; ***Weeratunga et al., 2020***). Within these endosomes, the cell determines which cargoes proceed to lysosomal degradation and which are rescued from degradation through recycling (***Simonetti and Cullen, 2019***; ***Cullen and Steinberg, 2018***; ***Yong et al., 2023***; ***Wang et al., 2018***). Cargoes retrieved from endosomes are packaged into tubulo-vesicular transport carriers and are either recycled back to the plasma membrane or routed to the *trans*-Golgi network (TGN) for re-entry into the exocytic pathway (***Simonetti and Cullen, 2019***; ***Cullen and Steinberg, 2018***; ***Yong et al., 2023***; ***Wang et al., 2018***).

Two central players in endosomal retrieval are Retromer and Commander. Retromer, a heterotrimeric complex comprised of VPS35, VPS29, and VPS26, recruits cargo proteins on the endosome and mediates the formation of tubulo-vesicular carriers for retrograde trafficking to the TGN or recycling to the plasma membrane (***Wang et al., 2018***; ***Rothman and Stevens, 1986***; ***Bankaitis et al., 1986***; ***Seaman et al., 1998***). The Commander complex, which primarily sorts cargoes from endosomal compartments to the plasma membrane, is a large protein complex composed of the Retriever subcomplex and the

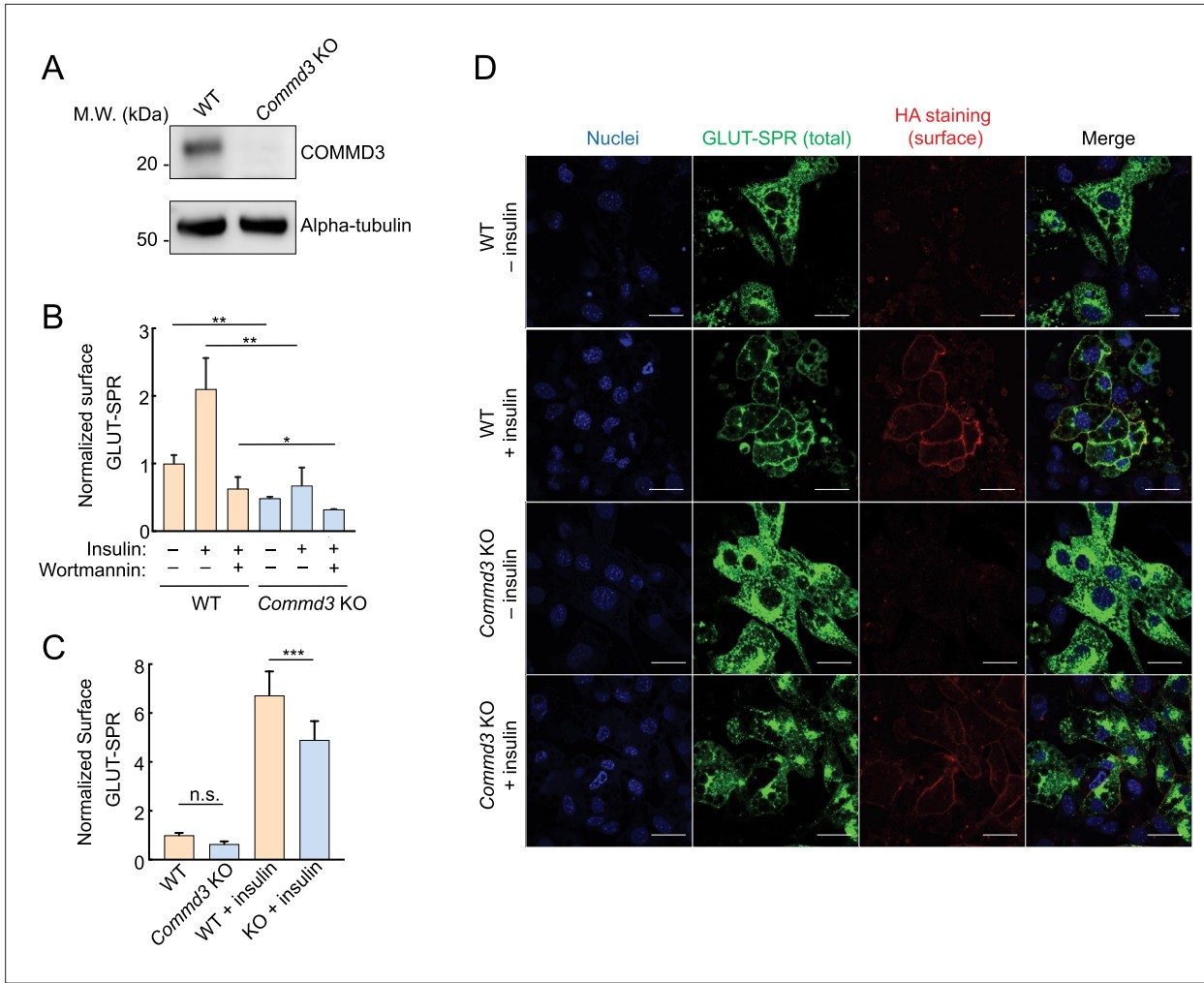

**Figure 2.** Intracellular sequestration of GLUT-SPR in COMMD3-deficient cells. (**A**) Representative immunoblots showing the indicated proteins in wild-type (WT) and *Commd3* KO mouse preadipocytes. (**B**) Normalized surface levels of GLUT-SPR measured by flow cytometry in WT and KO preadipocytes. The cells were either untreated or treated with 100 nM insulin for 1 hr before surface GLUT-SPR was labeled using anti-HA antibodies and APC-conjugated secondary antibodies. To calculate surface levels of GLUT-SPR, mean APC values were divided by mean GFP fluorescence. To inhibit insulin signaling, 100 nM wortmannin was added prior to insulin stimulation. In all figures, data normalization was performed by setting the mean value of WT data points as 100 or 1, and all data points including WT ones were normalized to that mean value. Data are presented as mean ± SD of three biological replicates. **p<0.01; *p<0.05 (calculated using Student's t-test). (**C**) Normalized surface levels of GLUT-SPR in WT and KO adipocytes. Data are presented as mean ± SD of three biological replicates. ***p<0.001; n.s., p>0.05 (calculated using Student's t-test). (**D**) Representative confocal images showing the localization of GLUT-SPR in unpermeabilized WT and *Commd3* KO adipocytes, which were either untreated or treated with 100 nM insulin for one hour. Surface GLUT-SPR was labeled using anti-HA antibodies and Alexa Fluor 568-conjugated secondary antibodies. Nuclei were stained with Hoechst 33342. Scale bars: 10 μm.

The online version of this article includes the following source data for figure 2:

**Source data 1.** PDF file containing original immunoblots for *Figure 2A*, indicating the relevant bands and treatments.

**Source data 2.** Original files for immunoblot analysis displayed in *Figure 2A*.

**Source data 3.** Flow cytometry data of wild-type (WT) and *Commd3* KO preadipocytes shown in *Figure 2B*.

**Source data 4.** Flow cytometry data of wild-type (WT) and *Commd3* KO adipocytes shown in *Figure 2C*.

CCC subcomplex (COMMD-CCDC22-CCDC93) (*Figure 1A*; *Healy et al., 2023*; *Boesch et al., 2024*; *Yong et al., 2023*; *Laulumaa et al., 2024*; *Leneva and Kovtun, 2024*; *Mallam and Marcotte, 2017*; *McNally et al., 2017*). The Retriever subcomplex is a heterotrimer consisting of VPS35L/C16ORF62, VPS26C/DSCR3, and VPS29 and shares a similar overall configuration with Retromer (*McNally et al., 2017*). The CCC complex includes 10 COMMD proteins (COMMD1-10), which interact through their conserved C-terminal copper metabolism MURR1 domains (COMMDs) (*Singla et al., 2019*; *van De*

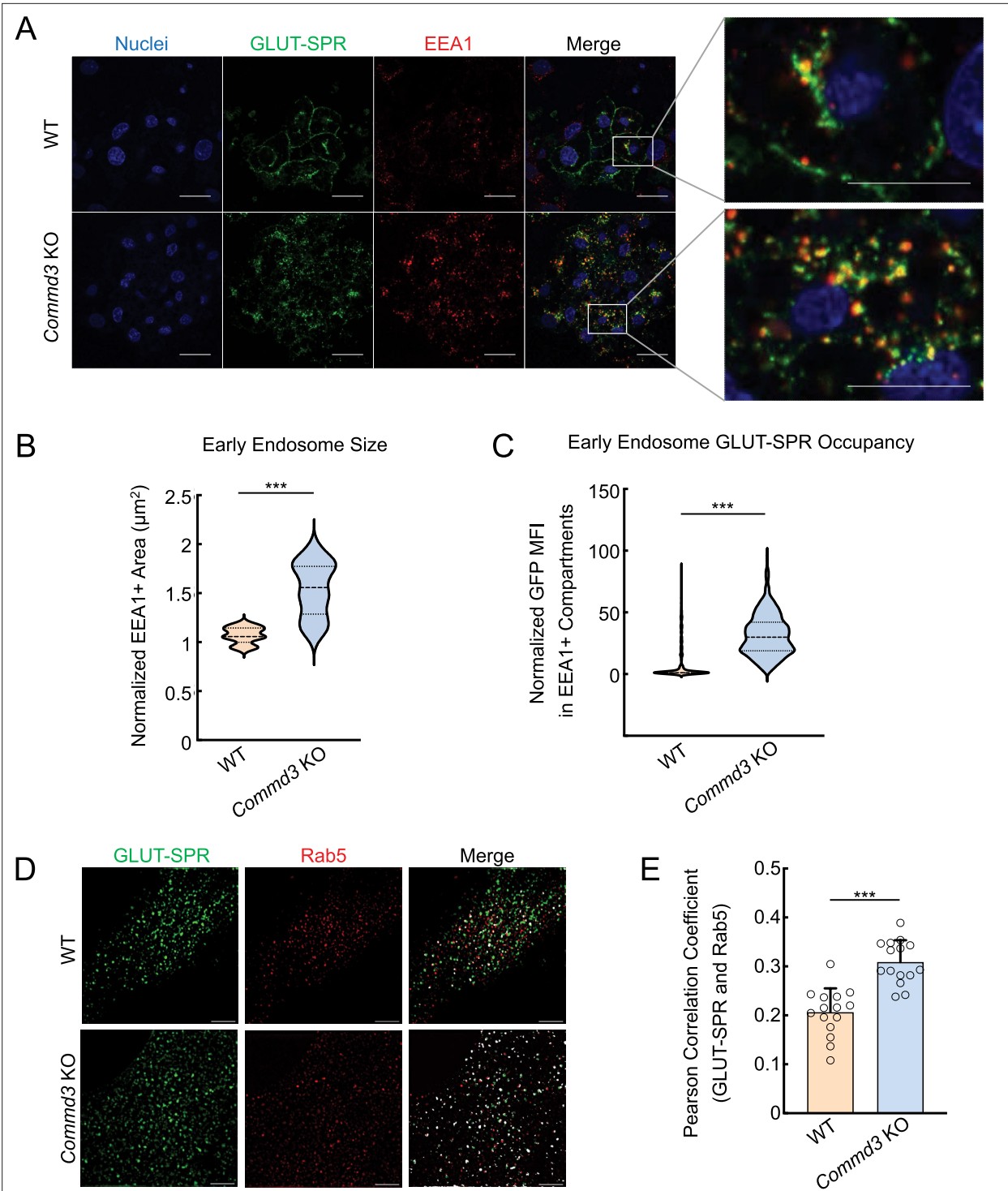

**Figure 3.** Abnormal endosomal morphology in COMMD3-deficient cells. (**A**) Representative confocal images showing the localization of EEA1 and GLUT-SPR in permeabilized wild-type (WT) and *Commd3* KO adipocytes stimulated with 100 nM insulin for 1 hr (scale bars: 10 μm). Enlarged inset images depict co-localization of EEA1 and GLUT-SPR (scale bars: 5 μm). (**B**) Violin plot showing quantification of EEA1-positive puncta using Fiji threshold analysis. Data of the KO adipocytes were normalized to those of WT cells. Three independent experiments are shown with ten cells analyzed in each experiment. ***p<0.001 (calculated using Student's t-test). (**C**) Violin plot depicting quantification of GFP mean fluorescence intensity (MFI) in EEA1-positive puncta using Fiji threshold analysis. Data of KO adipocytes were normalized to those of WT cells. Three independent experiments are shown with ten cells analyzed in each experiment. ***p<0.001 (calculated using Student's t-test). (**D**) Representative Structured Illumination Microscopy (SIM) images showing the subcellular localization of Rab5 and GLUT-SPR in WT and *Commd3* KO preadipocytes (scale bars: 5 μm). (**E**) Quantification

*Figure 3 continued on next page*

*Figure 3 continued*

of Rab5 and GLUT-SPR co-localization based on SIM images, which were captured as in (**D**) and analyzed using ImageJ. Each dot represents data of a subcellular region of interest. Five cells were analyzed and three regions per cell were quantified. ***p<0.001 (calculated using Student's t-test).

The online version of this article includes the following source data for figure 3:

**Source data 1.** Quantification of EEA1 + areas in wild-type (WT) and *Commd3* KO cells is shown in *Figure 3B*.

**Source data 2.** Quantification of GFP in EEA1 + compartments in wild-type (WT) and *Commd3* KO cells is shown in *Figure 3C*.

**Source data 3.** Flow cytometry data of wild-type (WT) and *Commd3* KO cells is shown in *Figure 3D*.

*Sluis et al., 2002*; *Burstein et al., 2005*). The COMMD proteins bind CCDC22 and CCDC93 to form the CCC subcomplex, which also contains DENND10 (*Healy et al., 2023*; *Leneva and Kovtun, 2024*). The 13-subunit CCC subcomplex combines with the Retriever subcomplex to form the 16-subunit Commander holo-complex (*Figure 1A*; *Healy et al., 2023*; *Boesch et al., 2024*; *Laulumaa et al., 2024*). On the endosome, Retromer and Commander cooperate with other factors such as sorting nexins (SNXs), Rab GTPases, actin, and the actin-remodeling WASH complex to capture cargoes and facilitate the formation of tubulo-vesicular carriers (*Simonetti and Cullen, 2019*; *Yong et al., 2023*; *Leneva and Kovtun, 2024*; *Yong et al., 2021*).

The Commander complex is ubiquitously expressed, and its 16 subunits display a stringent equimolar stoichiometry within the holo-complex (*Healy et al., 2023*; *Boesch et al., 2024*; *Laulumaa et al., 2024*; *Uhlén et al., 2015*), supporting the notion that these subunits function collectively in endosomal recycling. However, Commander subunits exhibit distinct tissue-specific expression patterns and can associate with different sets of proteins (*Laulumaa et al., 2024*; *Singla et al., 2019*; *Burstein et al., 2005*; *You et al., 2023*). Furthermore, CCC and Retriever also exist as subcomplexes, whereas COMMD proteins are found in pools of homo- and hetero-oligomers independent of the Commander complex (*Healy et al., 2023*; *Boesch et al., 2024*; *Singla et al., 2019*; *Healy et al., 2018*). These observations have led to the hypothesis that Commander subunits may have functions beyond their role in the Commander holo-complex (*Shirai et al., 2023*; *Nakai et al., 2019*; *Campion et al., 2018*; *Suraweera et al., 2021*). However, direct evidence for this hypothesis is still lacking.

In this work, we systematically dissected the Commander complex through unbiased genetic screens and comparative targeted mutations. Interestingly, we discovered that COMMD3, a subunit of the Commander complex, functions in endosomal recycling independently of other Commander subunits, in addition to its Commander-dependent activity. Comparative genetic analyses revealed that COMMD3 regulates a group of cargo proteins that do not require the Commander holo-complex. This Commander-independent function is mediated by the N-terminal domain (NTD) of COMMD3, which binds and stabilizes ADP-ribosylation factor 1 (ARF1). Guided by an AlphaFold-predicted structural model, we introduced point mutations into COMMD3 to disrupt its binding to ARF1. We found that these mutations diminish ARF1 expression and impair the Commander-independent function of COMMD3. Together, these findings uncovered a role of COMMD3 in endosomal trafficking independent of the Commander holo-complex, and suggest that other membrane trafficking complexes may also possess functions beyond their canonical roles within their respective complexes.

## Results

### Genetic analysis of the Retromer and Commander complexes using unbiased genome-wide CRISPR screens

Unbiased genome-wide genetic screens are a powerful approach to study the functions of membrane trafficking genes in cultured mammalian cells (*Wang et al., 2023*; *Gulbranson et al., 2019*; *Gulbranson et al., 2017*). Previously, we conducted a genome-wide CRISPR screen to identify new regulators of surface protein homeostasis using a surface protein reporter (GLUT-SPR) based on the glucose transporter GLUT4 (*Wang et al., 2023*). Mouse preadipocytes expressing the GLUT-SPR reporter were mutagenized using the genome-wide GeCKO v2 CRISPR library (*Ran et al., 2013*). In the genome-wide CRISPR screen, we used fluorescence-activated cell sorting (FACS) to isolate mutant preadipocytes with reduced surface levels of the GLUT-SPR reporter (*Wang et al., 2023*). The screen recovered known regulators of cargo exocytosis and enabled us to identify previously uncharacterized exocytic regulators, including Reps1 and Ralbp1 (*Wang et al., 2023*). Importantly, the new exocytic

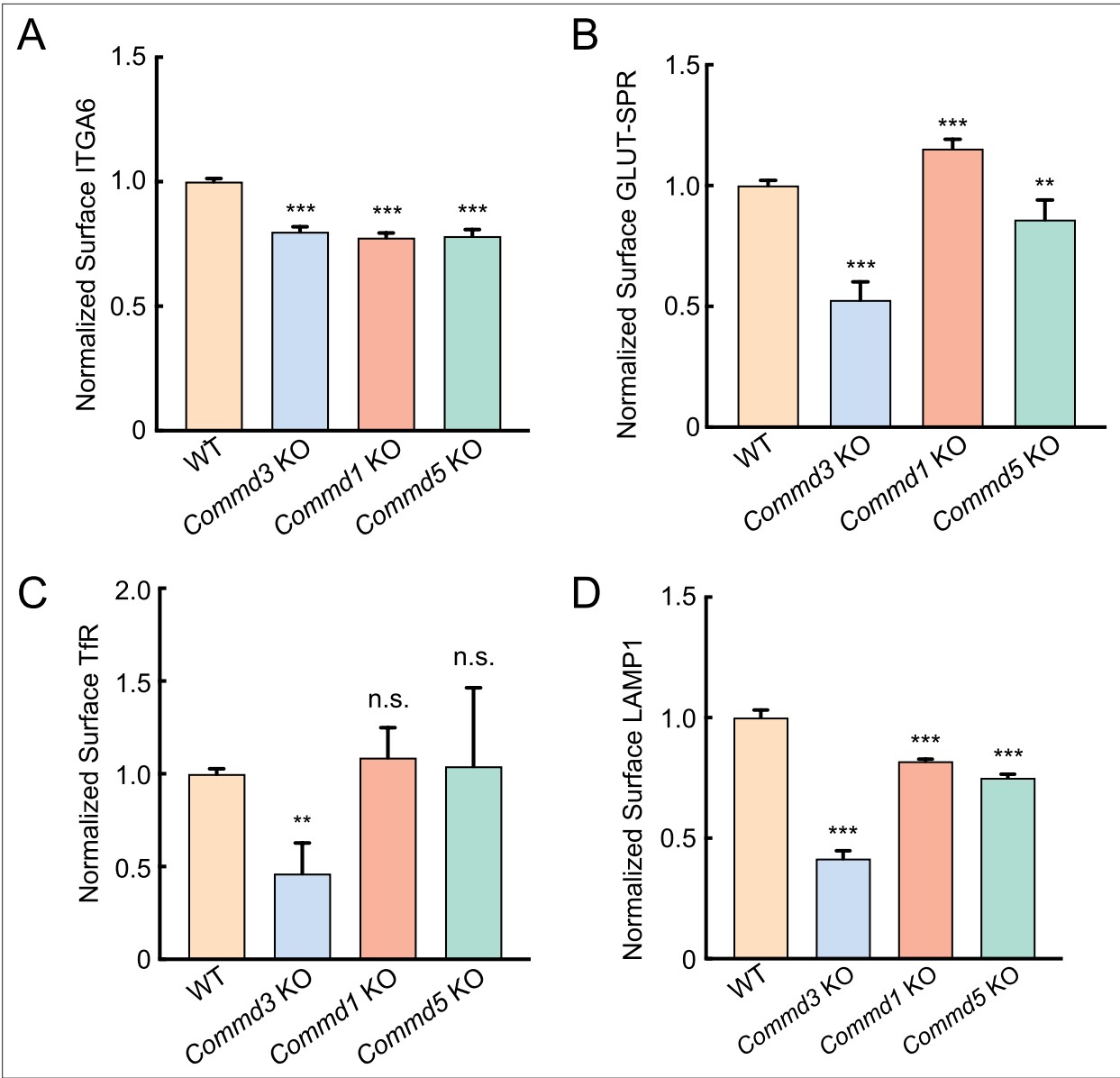

**Figure 4.** COMMD3 regulates a group of cargo proteins independent of other COMMD proteins. (**A–D**) Normalized surface levels of ITGA6 (**A**), GLUT-SPR (**B**), TfR (**C**), and LAMP1 (**D**) measured by flow cytometry in the indicated preadipocyte cell lines. To calculate surface levels of GLUT-SPR, mean APC values were divided by mean GFP fluorescence. Data of all cell samples were normalized to those of wild-type (WT) cells. Data of ITGA6 (n=3), GLUT-SPR (n=6), TfR (n=10), and LAMP1 (n=3) are presented as mean ± SD. **p<0.01; ***p<0.001; n.s., p>0.05 (calculated using one-way ANOVA).

The online version of this article includes the following source data and figure supplement(s) for figure 4:

**Source data 1.** Flow cytometry data of wild-type (WT) and KO cells is shown in *Figure 4A*.

**Source data 2.** Flow cytometry data of wild-type (WT) and KO cells is shown in *Figure 4B*.

**Source data 3.** Flow cytometry data of wild-type (WT) and KO cells is shown in *Figure 4C*.

**Source data 4.** Flow cytometry data of wild-type (WT) and KO cells are shown in *Figure 4D*.

**Figure supplement 1.** Quantification of transferrin receptor (TfR) in wild-type (WT) and mutant cell lines.

**Figure supplement 1—source data 1.** Flow cytometry data of wild-type (WT) and KO cells is shown in *Figure 4—figure supplement 1A*.

**Figure supplement 1—source data 2.** Flow cytometry data of wild-type (WT) and KO cells is shown in *Figure 4—figure supplement 1B*.

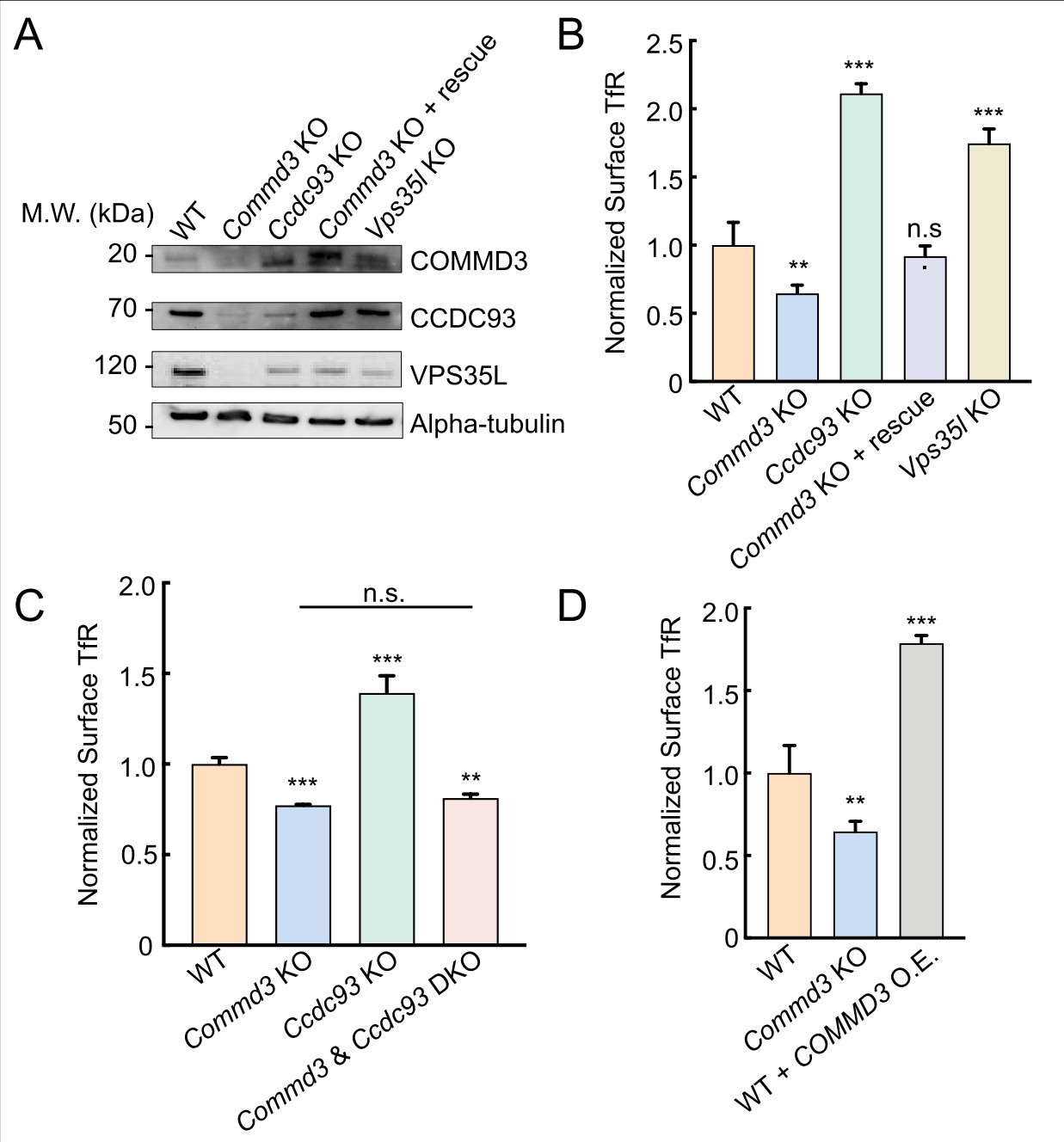

**Figure 5.** Upregulation of COMMD3 in cells deficient in CCDC93 or VPS35L. (**A**) Representative immunoblots showing protein expression in the indicated preadipocyte cell lines. (**B–D**) Normalized surface levels of transferrin receptor (TfR) measured by flow cytometry in the indicated preadipocyte cell lines. Data of all cell samples were normalized to those of WT cells. Data are presented as mean ± SD of three biological replicates. **p<0.01; ***p<0.001; n.s., p>0.05 (calculated using one-way ANOVA).

The online version of this article includes the following source data and figure supplement(s) for figure 5:

**Source data 1.** PDF file containing original immunoblots for *Figure 5A*, indicating the relevant bands and treatments.

**Source data 2.** Original files for immunoblot analysis displayed in *Figure 5A*.

**Source data 3.** Flow cytometry data of the indicated cell lines shown in *Figure 5B*.

**Source data 4.** Flow cytometry data of the indicated cell lines shown in *Figure 5C*.

**Source data 5.** Flow cytometry data of the indicated cell lines shown in *Figure 5D*.

**Figure supplement 1.** Quantification of total COMMD3 levels.

**Figure supplement 1—source data 1.** Flow cytometry data of the indicated cell lines is shown in *Figure 5—figure supplement 1*.

regulators identified in this GLUT-SPR-based CRISPR screen broadly regulate the surface proteostasis of membrane proteins (*Wang et al., 2023*).

To gain new insights into the molecular functions of Retromer and Commander, we re-analyzed the data from the CRISPR screen to examine genes encoding Retromer and Commander subunits. The CRISPR library used in the screen contained guide RNAs (gRNAs) targeting the genes encoding the subunits of the Retromer and Commander complexes (*Figure 1A and B*). The GLUT-SPR-based CRISPR screen isolated *Vps35* and *Vps29* (*Figure 1C*), which encode subunits of the Retromer complex, consistent with the critical role of Retromer in the endosomal recycling of GLUT4 (*Gulbranson et al., 2017*; *Pan et al., 2017*). Genes encoding Vps26 were not recovered in the screen due to redundancy (*Vps26a* and *Vps26b*). Recovery of the Retromer genes also suggests that the CRISPR screen has the sensitivity and specificity to systematically examine the functional roles of endosomal recycling genes. Genes encoding the unique subunits of the Retriever complex – *Vps35l* and *Vps26c* – were not isolated in the CRISPR screen (*Figure 1C*). Similarly, virtually none of the genes encoding the CCC complex were recovered as significant hits in the CRISPR screen, except for *Commd3*, which encodes the COMMD3 subunit of the Commander complex (*Figure 1A–C*; *Healy et al., 2023*). These data indicate that the Commander complex is dispensable for the endosomal recycling of GLUT-SPR.

To further characterize the functional roles of Retromer- and Commander-encoding genes in cell physiology, next we examined whether these genes are essential for cell viability or growth by re-analyzing data from an unbiased genome-wide essentiality screen. In the essentiality screen, mouse preadipocytes mutagenized by the CRISPR GeCKO v2 library were continuously passaged for two weeks without any selection (*Wang et al., 2023*). The abundance of gRNAs in this cell population was sequenced and compared with that in the original CRISPR gRNA library. If a gene is essential for cell viability or growth, its corresponding gRNAs would be depleted after the two-week cell passage. We found that none of the genes encoding the Retromer and Commander complexes are essential in mouse preadipocytes (*Figure 1D*). Thus, the disruption of either the Retromer or Commander-dependent endosomal retrieval pathways does not impair the viability or proliferation of these cells.

Together, these global genetic analyses suggest that COMMD3 plays an unrecognized role in GLUT-SPR trafficking, independent of its canonical function within the Commander holo-complex.

## Deletion of COMMD3 disrupts endosomal trafficking

To validate the role of COMMD3 in the surface homeostasis of GLUT-SPR, we deleted the *Commd3* gene in mouse preadipocytes (*Figure 2A*). Indeed, surface levels of GLUT-SPR were strongly reduced in *Commd3* KO preadipocytes, which were either cultured under standard conditions or stimulated with insulin to mimic a physiological fed state (*Figure 2B*). These data confirm the findings from the CRISPR screen (*Figure 1C*). To examine whether COMMD3 regulates the surface homeostasis of GLUT-SPR in another cell type, the fibroblast-like preadipocytes were differentiated into mature adipocytes, which are morphologically and functionally distinct from preadipocytes (*Ahfeldt et al., 2012*). Using flow cytometry, we observed that surface GLUT-SPR levels were markedly reduced in *Commd3* KO adipocytes (*Figure 2C*), indicating that the role of COMMD3 in surface protein homeostasis is not restricted to a single cell type. Similar findings were observed in adipocytes using confocal imaging (*Figure 2D*). Surface levels of GLUT-SPR were substantially decreased in *Commd3* KO cells, concomitant with elevated intracellular accumulation of the reporter (*Figure 2D*). These results demonstrated a critical role of COMMD3 in the endosomal trafficking of GLUT-SPR.

To further characterize the function of COMMD3 in the endosomal trafficking of GLUT-SPR, we stained for EEA1, an early endosome marker (*Mishra et al., 2010*), and examined the morphology of EEA1-positive organelles. We observed that the morphology of EEA1-positive endosomes was significantly altered in *Commd3* KO cells (*Figure 3A*). The size of the EEA1-positive endosomes markedly increased in *Commd3* KO cells (*Figure 3A and B*), reminiscent of the endosome enlargement observed in Retromer-deficient cells (*Neuman et al., 2021*). We also observed elevated GLUT-SPR accumulation in EEA1-positive endosomes in *Commd3* KO cells (*Figure 3C*). Next, we used Structured Illumination Microscopy (SIM) to visualize the localization of GLUT-SPR and Rab5, another marker of the early endosome (*Langemeyer et al., 2018*). We found that a portion of GLUT-SPR resided in Rab5-positive compartments and its co-localization with Rab5 increased in *Commd3* KO cells (*Figure 3D and E*).

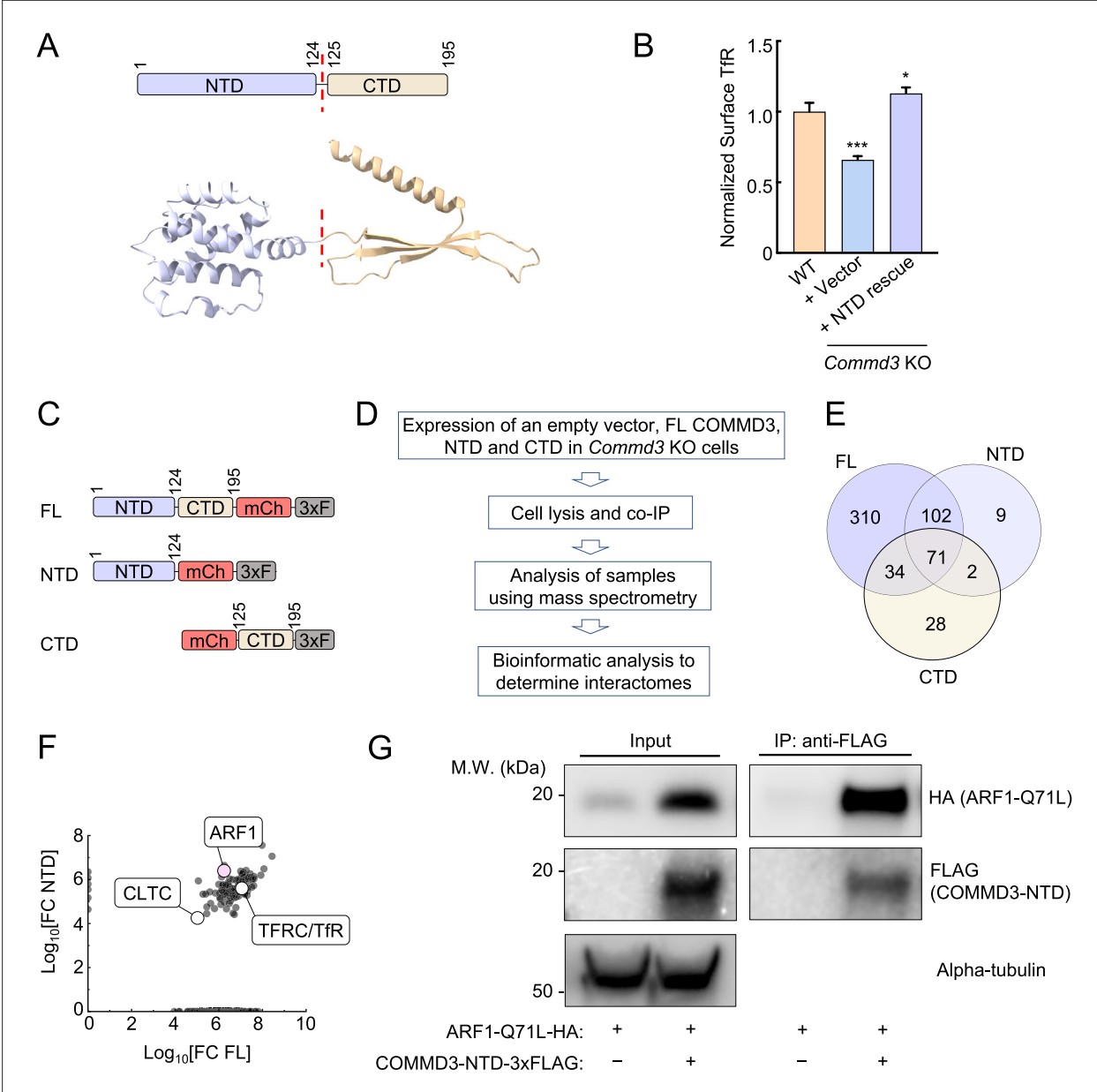

**Figure 6.** The Commander-independent function of COMMD3 is mediated by its N-terminal domain (NTD). (**A**) Top, diagram depicting the domain organization of COMMD3. Bottom: the structural model of COMMD3 (AlphaFold Protein Structure Database: AF-Q9UBI1-F1) showing the independently folded NTD and C-terminal domain (CTD). (**B**) Normalized surface levels of transferrin receptor (TfR) measured by flow cytometry in the indicated preadipocyte cell lines. Data of all cell samples were normalized to those of wild-type (WT) cells. Data are presented as mean ± SD of three biological replicates. *p<0.05; ***p<0.001 (calculated using one-way ANOVA). (**C**) Diagrams of full-length (FL) and truncated COMMD3 proteins used in proteomic experiments. The proteins were tagged with mCherry (mCh) and 3xFLAG (3xF). (**D**) Procedures of proteomic analysis to determine the interactomes of FL and truncated COMMD3 proteins. (**E**) Venn diagram showing the interactomes of COMMD3 proteins. (**F**) A scatter plot showing the fold change of protein abundance over vector control in the interactomes of FL COMMD3 and the NTD of COMMD3. Selected proteins are labeled. (**G**) Representative immunoblots showing the interactions of ARF1 Q71L with COMMD3-NTD. HA-tagged ARF1 Q71L and 3xFLAG-tagged COMMD3-NTD were co-expressed in 293T cells. COMMD3-NTD and associated proteins were immunoprecipitated using anti-FLAG antibodies and detected using immunoblotting. The ARF Q71L mutant was used here because it adopts a GTP-bound configuration (**Cohen and Donaldson, 2010**; **Zhang et al., 1994**).

The online version of this article includes the following source data and figure supplement(s) for figure 6:

**Source data 1.** Flow cytometry data of the indicated cell lines is shown in **Figure 6B**.

**Source data 2.** Original data of the plot is shown in **Figure 6F**.

*Figure 6 continued on next page*

*Figure 6 continued*

**Source data 3.** PDF file containing original immunoblots for *Figure 6G*, indicating the relevant bands and treatments.

**Source data 4.** Original files for immunoblot analysis displayed in *Figure 6G*.

**Figure supplement 1.** The C-terminal domain (CTD) of COMMD3 is unable to rescue transferrin receptor (TfR) surface levels.

**Figure supplement 1—source data 1.** Flow cytometry data of the indicated cell lines is shown in *Figure 6—figure supplement 1*.

Together, these data demonstrate that deletion of *Commd3* leads to the enlargement of early endosomes and accumulation of GLUT-SPR in these compartments, consistent with a trafficking defect occurring at the endosomal recycling step.

## COMMD3 regulates endosomal trafficking in a Commander-independent manner

Next, we sought to confirm the Commander-independent function of COMMD3 using comparative targeted mutations. We deleted genes encoding other Commander subunits and compared the phenotypes of the KO cells with that of *Commd3* KO cells. The COMMD proteins are found in three subcomplexes prior to forming the heterodecameric COMMD complex: subcomplex A (COMMD1-4-6-8), subcomplex B (COMMD2-3-4-8), and subcomplex C (COMMD5-7-9-10) (*Healy et al., 2023*). Here, we selected COMMD1, COMMD3, and COMMD5 as representative subunits of the three subcomplexes. We measured surface levels of integrin alpha-6 (ITGA6), a known Commander-dependent cargo (*McNally et al., 2017*), in cells lacking one of the COMMD proteins. As expected, surface levels of ITGA6 were downregulated in all these KO cell lines (*Figure 4A*), confirming the canonical role of COMMD3 within the Commander holo-complex.

Next, we examined the trafficking of other cargo proteins. We found that surface levels of GLUT-SPR were slightly increased in the *Commd1* KO cells, in contrast to the strong reduction observed in *Commd3* KO cells (*Figure 4B*). Surface levels of GLUT-SPR were only slightly affected in *Commd5* KO cells (*Figure 4B*). Surface levels of the transferrin receptor (TfR), a Commander-independent cargo (*Puthenveedu et al., 2010*), were markedly decreased in *Commd3* KO cells but remained intact in *Commd1* or *Commd5* KO populations (*Figure 4C*). Total TfR levels also decreased in *Commd3* KO cells (*Figure 4—figure supplement 1*), consistent with the notion that unretrieved cargo is routed to the lysosome for degradation (*Healy et al., 2023*; *Simonetti and Cullen, 2019*; *Cullen and Steinberg, 2018*). Surface levels of lysosomal-associated membrane protein 1 (LAMP1) were also strongly decreased in *Commd3* KO cells but were not substantially changed in *Commd1* or *Commd5* KO cells (*Figure 4D*). Together, comparative analysis of these diverse membrane proteins clearly demonstrates that COMMD3 regulates a group of cargo proteins independent of the Commander holo-complex, in addition to its canonical function within the Commander complex.

## Mutations of CCDC93 or Retriever lead to upregulation of COMMD3

To further characterize the Commander-independent function of COMMD3, we examined the KO phenotypes of other Commander subunits. We observed that the expression of CCDC93 and VPS35L was substantially decreased in *Commd3* KO cells, and VPS35L expression was diminished in *Ccdc93* KO cells. These findings are consistent with the notion that stability of subunits within a protein complex is interdependent (*Wang et al., 2023*; *Gulbranson et al., 2019*). Interestingly, we observed that COMMD3 expression was not reduced but upregulated in *Ccdc93* or *Vps35l* KO cells (*Figure 5A*, *Figure 5—figure supplement 1*). Thus, unlike other Commander subunits, COMMD3 persists when other subunits of the Commander complex are depleted, further supporting the ability of COMMD3 to regulate endosomal trafficking independent of the Commander holo-complex.

The upregulation of COMMD3 in Commander-deficient cells, which likely reflects a compensatory mechanism, offers another strategy to test the Commander-independent function of COMMD3. We reasoned that if a cargo is dependent on COMMD3 but not on the Commander complex, its surface levels could be increased in cells with elevated COMMD3. Indeed, we observed that surface levels of TfR, a cargo dependent on Commd3 but not on the Commander complex (*Figure 4C*; *Puthenveedu et al., 2010*), were markedly increased in *Ccdc93* or *Vps35l* KO cells (*Figure 5B*). Conversely, surface TfR levels were diminished in *Commd3* KO cells and were fully restored by expression of a *Commd3* rescue gene (*Figure 5B*). Total TfR expression was also reduced in *Commd3* KO cells and was rescued

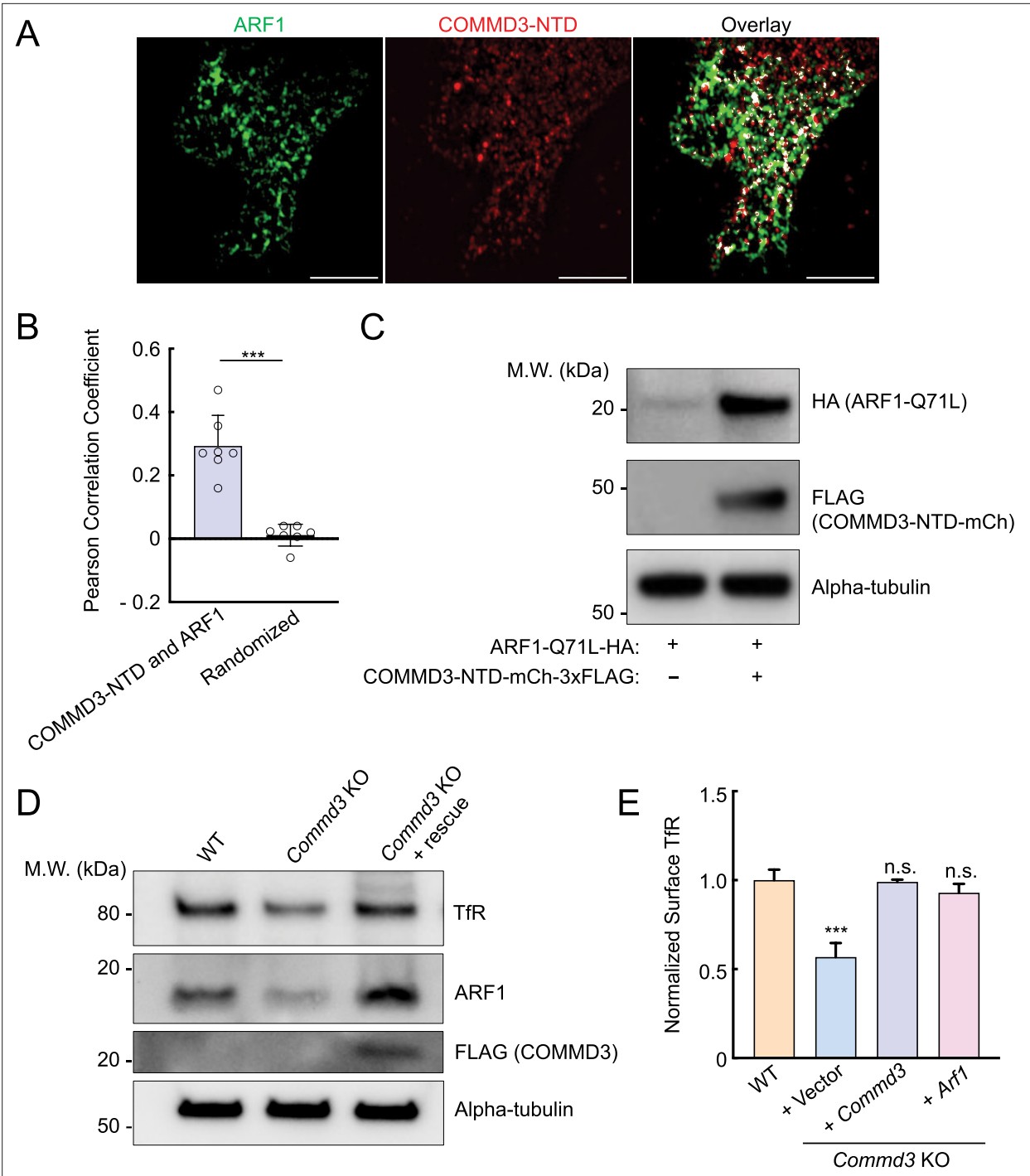

**Figure 7.** The N-terminal domain (NTD) of COMMD3 stabilizes ARF1. (**A**) Representative Structured Illumination Microscopy (SIM) images showing the subcellular localization of ARF1 and COMMD3-NTD expressed in HeLa cells. HA-tagged ARF1 and 3xFLAG-tagged COMMD3-NTD were transiently expressed in HeLa cells and stained using anti-HA and anti-FLAG antibodies, respectively (scale bars: 5 µm). (**B**) Quantification of ARF1 and COMMD3-NTD co-localization based on SIM images, which were captured as in (**A**) and analyzed using ImageJ. Each dot represents data of an individual cell. In randomized samples, ARF1 images were rotated 90° clockwise, whereas COMMD3-NTD images were not rotated. ***p<0.001 (calculated using Student's t-test). (**C**) Representative immunoblots showing the ARF1-stabilizing effects of COMMD3-NTD. HA-tagged ARF1 Q71L and 3xFLAG-tagged COMMD3-NTD were transiently expressed in HeLa cells and their total expression levels were measured using immunoblotting. (**D**) Representative immunoblots showing protein expression in the indicated preadipocyte cell lines. (**E**) Normalized surface levels of TfR measured by flow cytometry in the indicated preadipocyte cell lines. Data of all cell samples were normalized to those of wild-type (WT) preadipocytes. Data are presented as mean ± SD of three biological replicates. ***p<0.001; n.s., p>0.05 (calculated using one-way ANOVA).

*Figure 7 continued on next page*

*Figure 7 continued*

The online version of this article includes the following source data for figure 7:

**Source data 1.** Original data of the plot shown in *Figure 7B*.

**Source data 2.** PDF file containing original immunoblots for *Figure 7C and D*, indicating the relevant bands and treatments.

**Source data 3.** Original files for immunoblot analysis displayed in *Figure 7C and D*.

**Source data 4.** Flow cytometry data of the indicated cell lines is shown in *Figure 7E*.

by the *Commd3* rescue gene (Figure 7D, *Figure 4—figure supplement 1A*). KO of both *Commd3* and *Ccdc93* resulted in decreased surface levels of TfR (*Figure 5C*), confirming that the elevated surface TfR observed in *Ccdc93* KO cells was caused by COMMD3 upregulation. Consistent with this notion, overexpression of *Commd3* in WT cells markedly elevated TfR surface levels (*Figure 5D*). Altogether, these results demonstrate that COMMD3 is upregulated in Retriever- or CCDC93-deficient cells and enhances the endosomal retrieval of COMMD3-dependent cargoes, further supporting a Commander-independent role of COMMD3 in endosomal recycling.

## The NTD of COMMD3 mediates its Commander-independent function and interacts with ARF1

Next, we sought to determine the molecular mechanism by which COMMD3 regulates endosomal trafficking in a Commander-independent manner. The COMMD3 protein is comprised of two autonomously folded domains – NTD and CTD (*Figure 6A*). The C-terminal COMMD domains of COMMD proteins are well-conserved and mediate the formation of the heterodecameric COMMD complex within the Commander holo-complex (*Boesch et al., 2024*; *Maine and Burstein, 2007*). The NTDs of COMMD proteins share a similar structure but are more divergent in sequence (*Laulumaa et al., 2024*; *Healy et al., 2018*). The function of the COMMD NTDs is unknown. Next, we examined whether the Commander-independent function of COMMD3 relies on its NTD. We expressed the COMMD3 NTD in *Commd3* KO cells and examined whether the KO phenotype could be rescued. Indeed, expression of the COMMD3 NTD fully restored the surface levels of TfR in *Commd3* KO cells while the CTD did not (*Figure 6B*, *Figure 6—figure supplement 1*), indicating that the NTD is sufficient for this function.

To gain molecular insights into the NTD of COMMD3, we determined the interactomes of full-length (FL) COMMD3, NTD, and C-terminal domain (CTD) (*Figure 6C and D*). FL COMMD3, NTD, and CTD were individually expressed and isolated through co-immunoprecipitation (co-IP), and the associated proteins within these samples were identified using mass spectrometry. Through this proteomic analysis, we identified a group of proteins present in the interactomes of FL COMMD3 and NTD but absent from the CTD interactome (*Figure 6E*, *Supplementary file 1*). Among these proteins, ARF1 emerged as a strong candidate because it acts as a membrane trafficking regulator known to function on endosomes (*Figure 6F*; *Nakai et al., 2013*; *Stockhammer et al., 2024*). The ARF family of proteins is small GTPases involved in the recruitment of coat proteins, activation of membrane lipid-modifying enzymes, and interaction with the cytoskeleton (*Stockhammer et al., 2024*; *D'Souza-Schorey and Chavrier, 2006*). Using co-IP, we observed that the NTD of COMMD3 interacted with ARF1 (*Figure 6G*), confirming the results of the proteomic experiments. Together, these data demonstrate that the NTD of COMMD3 mediates its Commander-independent function and interacts with ARF1.

## COMMD3 regulates the stability of ARF1

To further characterize the interaction between COMMD3 and ARF1, we used SIM to visualize their subcellular localization. We observed significant co-localization between ARF1 and the NTD of COMMD3 (*Figure 7A and B*), consistent with their interactions detected in proteomic and co-IP experiments (*Figure 6*). We noted that, in the co-IP assays using 293T cells, ARF1 expression levels were significantly elevated when co-expressed with the NTD of COMMD3 (*Figure 6G*). A similar ARF1-stabilizing effect was also observed in HeLa cells (*Figure 7C*). These findings raised the possibility that COMMD3 uses its NTD to bind and stabilize ARF1. To test this model, we examined endogenous ARF1 levels in *Commd3* KO cells. Interestingly, we found that ARF1 levels were markedly reduced in *Commd3* KO cells and were fully restored upon introduction of a *COMMD3* rescue gene (*Figure 7D*). To further investigate the functional link between COMMD3 and ARF1, we overexpressed ARF1 in

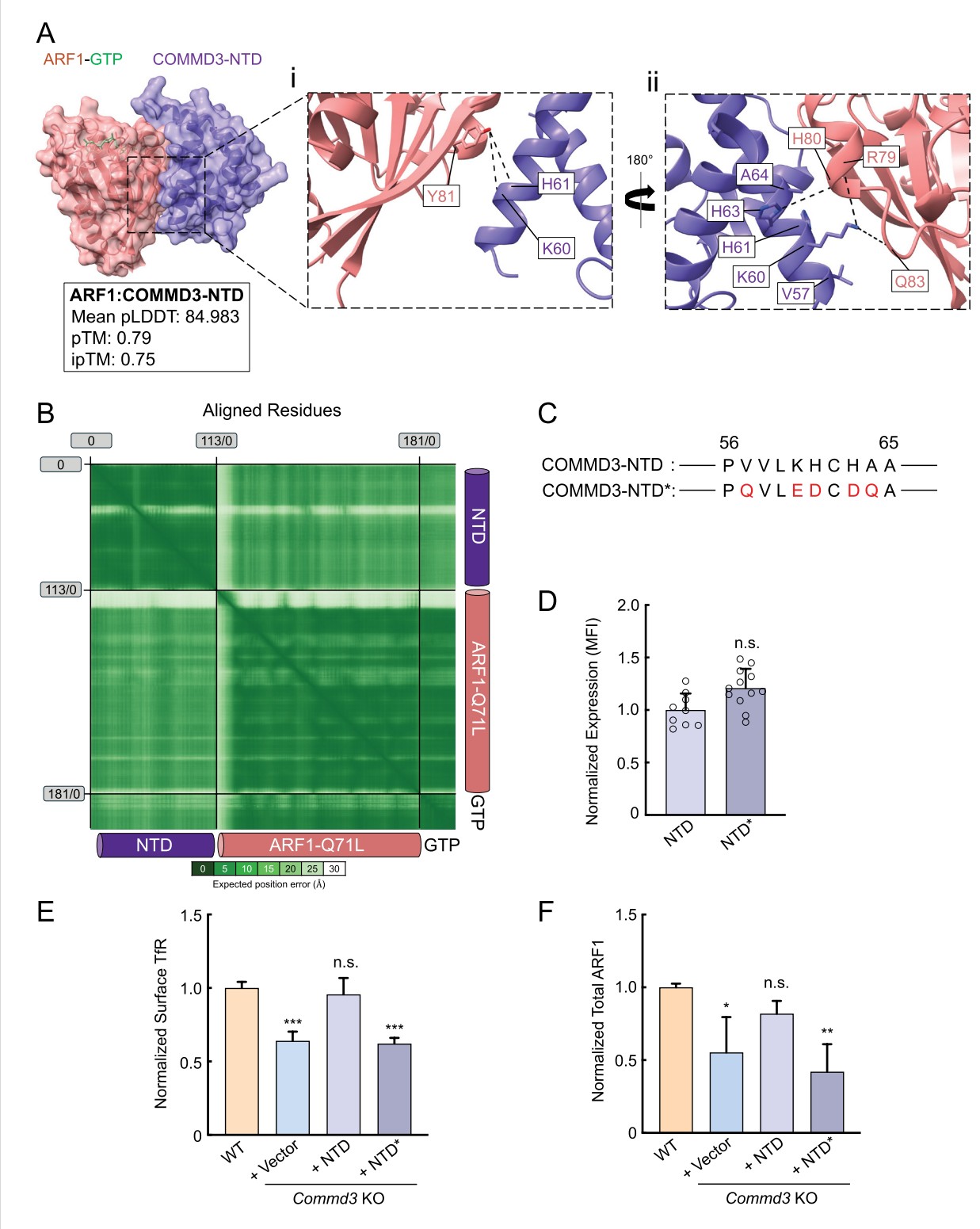

**Figure 8.** The COMMD3-ARF1 interaction is critical to the Commander-independent function of COMMD3. (**A**) Left: the AlphaFold3-predicted structure of the COMMD3:ARF1 heterodimer visualized using ChimeraX1.8. The structural prediction was performed using COMMD3-NTD (a.a. 1–120, purple), ARF1 (pink), and GTP (green) as input. Right: key residues at the COMMD3-ARF1 binding interface. The CIF file of the structural model is included in Supplementary Dataset 1. (**B**) Predicted alignment error (PAE) heatmap of the structural model shown in (**A**). (**C**) Diagrams showing the residues mutated in a COMMD3-NTD mutant (COMMD3-NTD*, only a.a. 56–65 are shown). Mutated residues are shown in red. (**D**) Quantification of COMMD-NTD and

*Figure 8 continued on next page*

*Figure 8 continued*

NTD* stably expressed in preadipocytes based on Structured Illumination Microscopy (SIM) images, which were captured and analyzed as in *Figure 7A–B*. Each dot represents data of an individual cell. MFI, mean fluorescence intensity. n.s., p>0.05 (calculated using Student's t-test). (**E**) Normalized surface levels of transferrin receptor (TfR) measured by flow cytometry in the indicated preadipocyte cell lines. Data of all cell samples were normalized to those of wild-type (WT) cells. Data are presented as mean ± SD of three biological replicates. ***p<0.001; n.s., p>0.05 (calculated using one-way ANOVA). (**F**) Quantification of endogenous ARF1 in the indicated preadipocyte cell lines based on intensities of proteins on immunoblots quantified using ImageJ. Data of all samples were normalized to those of WT cells. Data are presented as mean ± SD of three biological replicates. *p<0.05; n.s., p>0.05; **p<0.01 (calculated using one-way ANOVA).

The online version of this article includes the following source data for figure 8:

**Source data 1.** Flow cytometry data of the indicated cell lines shown in *Figure 8D*.

**Source data 2.** Flow cytometry data of the indicated cell lines shown in *Figure 8E*.

**Source data 3.** Flow cytometry data of the indicated cell lines shown in *Figure 8F*.

*Commd3* KO cells. Strikingly, ARF1 overexpression fully restored the surface levels of TfR in *Commd3* KO cells (*Figure 7E*). These data demonstrate that COMMD3 regulates endosomal trafficking through binding and stabilizing ARF1.

## Mutations disrupting the COMMD3-ARF1 interaction impair the Commander-independent function of COMMD3

Finally, we sought to determine whether ARF1 binding is required for the Commander-independent function of COMMD3. AlphaFold3 predicted a high-confidence structure of the ARF1-GTP:COMMD3-NTD heterodimeric complex (*Figure 8A and B*). According to this structural model, the α1 helix of COMMD3-NTD binds to the switch 1 of ARF1, while the α3 and α4 helices of COMMD3-NTD interact with the switch 2 of ARF1 (*Figure 8A and B*, *Source data 1*). The switches 1 and 2 of ARF1 are highly conserved regions that are regulated by GTP binding and interact with effectors involved in endosomal recycling (*Goldberg, 1998*; *Sauvageau et al., 2017*). The binding of COMMD3 to ARF1 is mainly mediated by hydrophobic interactions and hydrogen bonds, including hydrogen bonds formed by K60 and H61 of COMMD3 with Y81 of ARF1, K60 of COMMD3 with R79 and Q83 of ARF1, H63 of COMMD3 with H80 of ARF1 (*Figure 8A and B*, *Source data 1*). AphaFold3 did not predict high-confidence structural models between COMMD3-NTD and ARF1-GDP or apo-ARF1, suggesting that COMMD3 selectively recognizes the GTP-bound form of ARF1. This conclusion is consistent with the observation that COMMD3 strongly stabilizes GTP-bound ARF1 (*Figures 6G and 7C*).

Altogether, these results further support the conclusion that COMMD3 regulates endosomal trafficking through binding and stabilizing ARF1 (*Figure 9*).

Next, we introduced point mutations into the NTD of COMMD3 based on the structural model, aiming to disrupt its interaction with ARF1 (*Figure 8C*). The COMMD3-NTD mutant was expressed at a similar level as the WT protein in *Commd3* KO cells (*Figure 8D*). We observed that the COMMD3-NTD mutant failed to restore surface levels of TfR, whereas WT COMMD3-NTD fully rescued the KO phenotype (*Figure 8E*). In agreement with this finding, WT COMMD3-NTD, but not the mutant, restored ARF1 expression in *Commd3* KO cells (*Figure 8F*). Altogether, these results further support the conclusion that COMMD3 regulates endosomal trafficking through binding and stabilizing ARF1.

## Discussion

The genetic evidence for a Commander-independent function of COMMD3 in endosomal trafficking is threefold. First, COMMD3 was the only Commander subunit isolated as a significant hit in a genome-scale genetic screen dissecting the surface homeostasis of GLUT-SPR, revealing a unique role of COMMD3 among Commander subunits in the GLUT-SPR trafficking pathway. Unbiased genetic screens are particularly powerful in dissecting a protein complex because they systematically interrogate the functional roles of all subunits within the complex (*Wang et al., 2023*; *Gulbranson et al., 2019*; *Gulbranson et al., 2017*; *Menasche et al., 2020*; *Davis et al., 2015*). Second, our comparative targeted mutations confirmed that the loss-of-function phenotype of COMMD3 is fundamentally distinct from that of other COMMD proteins. Besides GLUT-SPR, a group of other cargoes including

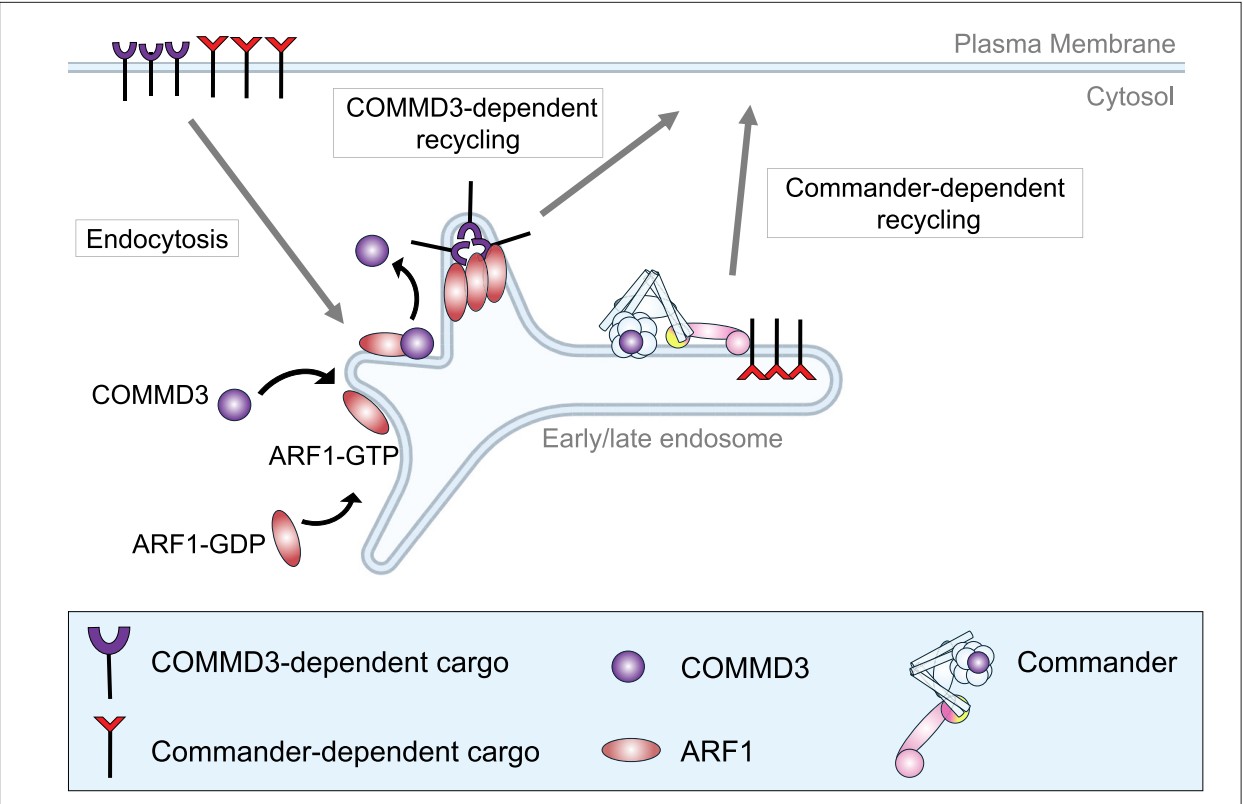

**Figure 9.** Model of the Commander-independent function of COMMD3 in endosomal trafficking. For clarity, Retromer and other endosomal recycling regulators are not shown.

TfR also selectively rely on COMMD3 but not on COMMD1 or COMMD5, further supporting the unique role of COMMD3 in the trafficking of these cargo proteins. It should be noted that our findings of ITGA6 are fully consistent with a canonical function of COMMD3 within the Commander holo-complex in endosomal recycling. Third, while the stability of Commander subunits is generally interdependent, COMMD3 persists when other subunits of the Commander complex are depleted. In fact, COMMD3 is upregulated when CCDC93 or Retriever are mutated, resulting in elevated surface levels of TfR.

The Commander-independent function of COMMD3 is mediated by its NTD, a domain previously lacking an ascribed function, whereas its C-terminal COMMD domain is required for the canonical role of COMMD3 within the Commander complex (*Healy et al., 2023*; *Boesch et al., 2024*; *Laulumaa et al., 2024*). At the molecular level, the NTD of COMMD3 binds and stabilizes ARF1, a member of the ARF small GTPase family playing a key role in endosomal recycling (*Nakai et al., 2013*; *Kondo et al., 2012*). These findings suggest that GTP-ARF1 is intrinsically unstable prior to engaging its effectors and, therefore, requires stabilization by COMMD3. Since COMMD3 binds to the same regions of ARF1 recognized by ARF1 effectors, it must dissociate from ARF1 before the latter can engage its effectors to regulate endosomal recycling (*Stockhammer et al., 2024*). Further research is needed to determine whether COMMD3 is released from ARF1 through direct competition by ARF1 effectors or if a more active mechanism is involved.

COMMD proteins are known to exist outside the Commander holo-complex. While they are unstable as monomers, COMMD proteins can form homo- or hetero-oligomers independently of other Commander subunits (*Healy et al., 2018*; *Shirai et al., 2023*; *Nakai et al., 2019*; *Li et al., 2015*). Since other COMMD proteins were not identified as significant hits in our GLUT-SPR-based CRISPR screen, COMMD3 likely forms homo-oligomers before associating with ARF1 to regulate cargo trafficking in a Commander-independent manner. The Commander-independent function of COMMD3 offers an additional route of endosomal recycling, alongside the Retromer- and Commander-dependent recycling pathways. By leveraging existing proteins such as COMMD3, the cell expands the repertoire

of endosomal recycling routes without increasing gene numbers, which helps meet the formidable challenge of sorting thousands of membrane proteins that are constantly endocytosed into the cell. An important future direction is to determine precisely how COMMD3 interacts with ARF1 and cooperates with other endosomal regulators, such as SNXs, to recognize sorting signals on COMMD3-dependent cargoes during endosomal retrieval.

Our findings raise the intriguing possibility that other COMMD proteins may also possess additional functions outside the Commander holo-complex. In addition to endosomal recycling, Commander is also implicated in biological pathways including vesicle fusion, cytoskeletal organization, intracellular signaling, protein degradation, and gene expression (*Ambrosio et al., 2022*; *McNally et al., 2017*; *Singla et al., 2019*; *Shirai et al., 2023*; *Nakai et al., 2019*; *Campion et al., 2018*; *Suraweera et al., 2021*; *Li et al., 2015*; *Bartuzi et al., 2013*; *Esposito et al., 2016*; *O'Hara et al., 2014*; *Riera-Romo, 2018*; *You et al., 2018*; *Mao et al., 2011*; *de Bie et al., 2006*; *Maine et al., 2007*; *Phillips-Krawczak et al., 2015*; *Bartuzi et al., 2016*; *Hancock et al., 2023*; *Liu et al., 2013*; *Weiskirchen and Penning, 2021*; *Jiang et al., 2019*; *Dumoulin et al., 2020*). Although many of these pathways are expected to require the Commander holo-complex, others may be mediated by stable pools of individual COMMD proteins outside the Commander complex. These Commander-dependent functions of COMMD proteins are supported by their evolutionary history. COMMD proteins evolved later than other Commander components, concurrent with the expansion of genes encoding membrane trafficking regulators (*Liebeskind et al., 2019*). It is possible that precursors of COMMD proteins may have been stable and functional prior to joining the more ancient Commander core complex. As the Commander holo-complex emerged, some of the primordial functions of COMMD precursors were likely retained in COMMD proteins. Further evidence supporting Commander-independent functions is the distinct tissue-specific expression patterns of COMMD proteins (*Laulumaa et al., 2024*; *Singla et al., 2019*; *Burstein et al., 2005*; *You et al., 2023*; *Yang et al., 2019*). Given their differential expression, a portion of COMMD proteins inevitably exists outside the Commander holo-complex and may play cell type-specific physiological roles. Besides Commander, other membrane trafficking complexes might also contain subunits functioning outside the homo-complexes. In line with this notion, SEC13, a component of the COPII coat formed on the endoplasmic reticulum (ER), is also implicated in the formation of the nuclear pore complex (NPC) (*Niu et al., 2014*; *Enninga et al., 2003*). We suggest that the genetic strategy described in this study will be instrumental in determining whether and how COMMD proteins and SEC13 regulate cell physiology independently of their respective holo-complexes.

# Materials and methods

**Key resources table**

| Reagent type (species) or resource | Designation | Source or reference | Identifiers | Additional information |
|---|---|---|---|---|
| Gene (*H. sapiens*) | COMMD3 | Uniprot | Q9UBI1 | Sequence codon optimized to avoid Cas9 targeting |
| Gene (*H. sapiens*) | ARF1 | Entrez | PVNH8 | Cloned from Addgene # 39554 |
| Cell line (*M. musculus*) | Preadipocytes | Dr. Shingo Kajimura | N/A | |
| Cell line (*H. sapiens*) | HeLa | ATCC | CCL-2 | |
| Cell line (*H. sapiens*) | HEK 293T | ATCC | CRL-3216 | |
| Recombinant DNA reagent (*M. musculus*) | pLenti-CRISPR-v2 | Addgene | #52961 | Lentiviral construct to infect and express Cas9 and gRNA. |
| Recombinant DNA reagent (*M. musculus*) | pLentiGuide-Puro vector | Addgene | #52963 | Lentiviral construct to infect and express gRNA. |
| Recombinant DNA reagent (*H. sapiens*, *M. musculus*) | SHC003 GFPD | Addgene | #133301 | Mammalian expression plasmid backbone. |

*Continued on next page*

*Continued*

| Reagent type (species) or resource | Designation | Source or reference | Identifiers | Additional information |
|---|---|---|---|---|
| Recombinant DNA reagent (*H. sapiens, M. musculus*) | SHC003 GFPD-humanCOMMD3-FL-mCherry-3xFLAG | This paper | N/A | Lentiviral construct to infect/transfect cells and express genes. |
| Recombinant DNA reagent (*H. sapiens, M. musculus*) | SHC003 GFPD-humanCOMMD3-NTD-mCherry-3xFLAG | This paper | N/A | Lentiviral construct to infect/transfect cells and express genes. |
| Recombinant DNA reagent (*H. sapiens, M. musculus*) | SHC003 GFPD-mCherry-humanCOMMD3-CTD-3xFLAG | This paper | N/A | Lentiviral construct to infect/transfect cells and express genes. |
| Recombinant DNA reagent (*H. sapiens, M. musculus*) | SHC003 GFPD-humanARF1-HA | This paper | N/A | Lentiviral construct to infect/transfect cells and express genes. |
| Sequence-based reagents | COMMD3 CRISPR guide RNA sequence primers | This paper | N/A | CTTCGCGCTTCTCCTCCGGG |
| Sequence-based reagents | COMMD3 CRISPR guide RNA sequence primers | This paper | | CTTGAAACAGATCGACCCAG |
| Sequence-based reagents | COMMD1 CRISPR guide RNA sequence primers | This paper | | TCACGGACACTCGGGTGTCA |
| Sequence-based reagents | Commd1 CRISPR guide RNA sequence primers | This paper | | ACTGCTCAAACCAAAAAGCA |
| Sequence-based reagents | Commd5 CRISPR guide RNA sequence primers | This paper | | GTTGTTGAAACTCGTAGTCG |
| Sequence-based reagents | Commd5 CRISPR guide RNA sequence primers | This paper | | TGCCAGCGCCAACCTGTCAG |
| Sequence-based reagents | Ccdc93 CRISPR guide RNA sequence primers | This paper | | CGAAAGTACCGACGGCAGCG |
| Sequence-based reagents | Ccdc93 CRISPR guide RNA sequence primers | This paper | | GATGACCGCCATGGCAAACG |
| Sequence-based reagents | Vps35l CRISPR guide RNA sequence primers | This paper | | GGATTATGTGAACCGCATAG |
| Sequence-based reagents | Vps35l CRISPR guide RNA sequence primers | This paper | | GGAGGTTTGCAAGTGCATCA |
| Antibody | anti-HA mouse monoclonal | BioLegend | #901501, RRID:AB_2565006 | Flow (1:250), IF (1:1000) |
| Antibody | APC-conjugated anti-LAMP1 mouse monoclonal | BioLegend | #328619, RRID:AB_1279055 | Flow (1:250) |
| Antibody | anti-ITGA6 rat monoclonal | Invitrogen | #14-0495-82, RRID:AB_891480 | Flow (1:250) |
| Antibody | APC-conjugated anti-mouse secondary antibodies rat monoclonal | eBioscience | #17-4015-82, RRID:AB_2573205 | Flow (1:1000) |
| Antibody | APC-conjugated anti-rat antibodies goat polyclonal | Invitrogen | #A10540, RRID:AB_10562535 | Flow (1:1000) |
| Antibody | APC-conjugated anti-TfR/CD71 antibodies mouse monoclonal | BioLegend | #334108, RRID:AB_10915138 | Flow (1:500) |
| Antibody | anti-FLAG M2 mouse monoclonal | Sigma-Aldrich | #F1804, RRID:AB_262044 | IF (1:1000) |
| Antibody | Alexa Fluor 647-conjugated anti-rabbit IgG goat polyclonal | Invitrogen | #A32733, RRID:AB_2633282 | IF (1:1000) |

*Continued on next page*

*Continued*

| Reagent type (species) or resource | Designation | Source or reference | Identifiers | Additional information |
|---|---|---|---|---|
| Antibody | Alexa Fluor 568-conjugated anti-mouse IgG goat polyclonal | Thermo Fisher Scientific | #A11004, RRID:AB_2534072 | IF (1:1000) |
| Antibody | Anti-COMMD3 rabbit polyclonal | Bethyl | #A304-092A, RRID:AB_2621341 | IB (1:200) |
| Antibody | anti-VPS35L rabbit polyclonal | Invitrogen | #PA5-28553, RRID:AB_2546029 | IB (1:500) |
| Antibody | anti-CCDC93 mouse monoclonal | Santa Cruz Biotechnology | #sc-514600 | IB (1:100) |
| Antibody | anti-alpha-tubulin mouse monoclonal | DSHB | #12G10, RRID:AB_1210456 | IB (1:1000) |
| Antibody | HRP-conjugated anti-FLAG M2 mouse monoclonal | Sigma-Aldrich | #A8592, RRID:AB_439702 | IB (1:1000) |
| Antibody | HRP-conjugated anti-HA rat monoclonal | Roche | #12013819001, RRID:AB_390917 | IB (1:1000) |
| Antibody | HRP-conjugated anti-rabbit IgG goat polyclonal | Sigma-Aldrich | #A6154, RRID:AB_258284 | IB (1:2000) |
| Antibody | HRP-conjugated anti-mouse IgG sheep polyclonal | Sigma-Aldrich | #A6782, RRID:AB_258315 | IB (1:2000) |
| Software, algorithm | AlphaFold3 (AF3) | PMID:38718835 | | Used for structural prediction |

## Cell lines and cell culture

HeLa and 293T cells were obtained from ATCC and had been authenticated by the vendor. Mouse preadipocytes were immortalized cells derived from mouse adipose tissue and were validated using expression markers such as GLUT4, as well as functional assays including differentiation into adipocytes and insulin responsiveness. All cell lines were routinely tested for mycoplasma contamination by the StemTech core facility at the University of Colorado Boulder. The cells were cultured in Dulbecco's Modified Eagle Medium (DMEM) supplemented with 10% FB Essence (FBE, VWR, #10803–034) and penicillin/streptomycin (Thermo Scientific, #15140122). The cells were maintained in a humidified incubator at 37 °C with 5% $CO_2$. To differentiate preadipocytes into mature adipocytes, preadipocytes were grown to ~95% confluence before a differentiation cocktail was added at the following final concentrations: 5 µg/mL insulin (Sigma, #I0516), 1 nM Triiodo-L-thyronine (T3, Sigma, #T2877), 125 µM indomethacin (Sigma, #I-7378), 5 µM dexamethasone (Sigma, #D1756), and 0.5 mM 3-isobutyl-1-methylxanthine (IBMX, Sigma, #I5879). After two days, the cells were switched to DMEM supplemented with 10% FBE, 5 µg/mL insulin, and 1 nM T3. After another two days, fresh DMEM media supplemented with 10% FBE and 1 nM T3 were added to the cells. Differentiated adipocytes were usually analyzed six days after addition of the differentiation cocktail.

## Gene KO using CRISPR-Cas9

Genome-wide CRISPR screens of GLUT-SPR surface homeostasis and gene essentiality were described previously (*Wang et al., 2023*). To individually ablate a candidate gene, gRNAs targeting the gene were chosen via the CRISPick algorithm (https://portals.broadinstitute.org/gppx/crispick/public) to maximize KO efficiency and minimize off-target effects. The upstream guide was cloned into the pLenti-CRISPR-v2 vector (Addgene, #52961) and the downstream guide was cloned into a modified version of the pLentiGuide-Puro vector (Addgene, #52963), in which the puromycin selection marker was replaced with a hygromycin selection marker.

Guide sequences targeting the mouse *Commd3* gene are: CTTCGCGCTTCTCCTCCGGG and CTTGAAACAGATCGACCCAG. Guide sequences targeting the mouse *Commd1* gene are: TCAC GGACACTCGGGTGTCA and ACTGCTCAAACCAAAAAGCA. Guide sequences targeting the mouse *Commd5* gene are: GTTGTTGAAACTCGTAGTCG and TGCCAGCGCCAACCTGTCAG. Guide sequences targeting the mouse *Ccdc93* gene are: CGAAAGTACCGACGGCAGCG and GATGACCG CCATGGCAAACG. Guide sequences targeting the mouse *Vps35l* gene are: GGATTATGTGAACCGC

ATAG and GGAGGTTTGCAAGTGCATCA. Double KO cells of *Commd3* and *Ccdc93* were generated using one guide per gene: *Commd3* – CTTCGCGCTTCTCCTCCGGG and *Ccdc93* – GATGACCG CCATGGCAAACG.

CRISPR plasmids were transfected into HEK 293T cells along with pAdVAntage (Promega, #E1711), pCMV-VSVG (Addgene, #8454), and psPax2 (Addgene, #12260). The 293T cell culture media containing lentiviral particles were harvested daily for four days and centrifuged at 25,000 rpm (113,000 g) for 1.5 hr at 4 °C using a Beckman SW28 rotor. Viral pellets were resuspended in PBS and used to infect target cells. After lentiviral infection, cells were selected using 3.5 µg/mL puromycin (Sigma, #3101118) for 2 days, followed by selection using 500 µg/mL hygromycin B (Thermo, #10687010) for another 2 days. All KO cells used in this work were pooled KO cell populations.

## Gene expression in mammalian cells

The codon-optimized human *COMMD3* gene with a 3xFLAG-encoding sequence or an mCherry-3xFlag-encoding sequence was subcloned into the SHC003BSD-GFPD vector (Addgene, #133301). This *COMMD3* gene was not targeted by gRNAs used in the KO experiments. The plasmid expressing the NTD (amino acids 1–124) of human COMMD3 was generated in a similar way. For the IP experiments, DNA fragments encoding FL and truncated human COMMD3 were subcloned into the SHC003BSD-GFPD vector. FL COMMD3 (amino acids 1–195) and NTD were fused to mCherry and 3xFlag tags at their C-termini. The CTD (amino acids 125–195) was fused to an mCherry tag at its N-terminus and a 3xFlag tag at its C-terminus. The human *ARF1* gene was subcloned into the SHC003BSD-GFPD vector with an HA-encoding sequence at the 3' end. The constructs were transfected into 293T cells to produce lentiviral particles using a similar procedure as CRISPR lentiviral production. The lentiviruses were used to infect target cells, followed by selection using 10 µg/mL blasticidin (Thermo Fisher Scientific, #BP2647).

## Flow cytometry

Cells grown on cell culture plates were washed with ice-cold KRH buffer and blocked at 4 °C with KRH buffer supplemented with 5% FBE. Subsequently, the cells were labeled for 1 hr with KRH buffer containing 2% FBE and the following antibodies: anti-HA (BioLegend, #901501, RRID:AB_2565006), APC-conjugated anti-LAMP1 (BioLegend, #328619, RRID:AB_1279055), anti-ITGA6 (Invitrogen, #14-0495-82, RRID:AB_891480), APC-conjugated anti-mouse secondary antibodies (eBioscience, #17-4015-82, RRID:AB_2573205), APC-conjugated anti-rat antibodies (Invitrogen, #A10540, RRID:AB_10562535), and APC-conjugated anti-TfR/CD71 antibodies (BioLegend, #334108, RRID:AB_10915138). Following antibody labeling, cells were washed twice with KRH buffer containing 5% FBE and once with PBS. Cells were dissociated using Accutase before resuspension in PBS buffer containing 5% FBE. Cells were analyzed in triplicate on a CyAN ADP analyzer (Beckman Coulter).

## Immunostaining and imaging

Cells grown on glass coverslips were washed with PBS and fixed using 4% PFA in PBS. Cells were then permeabilized using 0.1% Tween-20 and blocked with PBS buffer containing 5% FBE. Permeabilization was omitted when surface proteins were stained. Cells were labeled for 1 hr with 2% FBE in PBS using the following antibodies: anti-HA, anti-FLAG M2 (Sigma-Aldrich, #F1804, RRID:AB_262044), Alexa Fluor 647-conjugated anti-mouse IgG (Invitrogen, #A32733, RRID:AB_2633282), and Alexa Fluor 568-conjugated anti-rabbit IgG (Thermo Fisher Scientific, #A11004, RRID:AB_2534072). Confocal images were acquired on a Nikon A1 laser scanning confocal microscope using a ×100×oil immersion objective. In SIM, images were captured using a 100 x oil immersion objective on a Nikon SIM microscope as previously described (*Wan et al., 2024*).

## Immunoprecipitation (IP) and immunoblotting

In IP experiments, cells were lysed in IP buffer (25 mM HEPES [pH 7.4], 138 mM NaCl, 10 mM Na3PO4, 2.7 mM KCl, 0.5% CHAPS, 1 mM DTT, and a protease inhibitor cocktail). After centrifugation, proteins were immunoprecipitated from cell extracts using primary antibodies and protein A/G agarose beads (Thermo Scientific, #WF324079). For immunoblotting, immunoprecipitates or whole cell lysates were resolved on 8% Bis-Tris SDS–polyacrylamide gel electrophoresis (SDS-PAGE) and transferred to PVDF membranes. Proteins were detected using unlabeled primary antibodies and horseradish

peroxidase (HRP)-conjugated secondary antibodies, or HRP-conjugated primary antibodies. Primary antibodies used in immunoblotting include anti-COMMD3 (Bethyl, #A304-092A, RRID:AB_2621341), anti-VPS35L (Invitrogen, #PA5-28553, RRID:AB_2546029), anti-CCDC93 (Santa Cruz Biotechnology, #sc-514600), anti-alpha-tubulin (DSHB, #12G10, RRID:AB_1210456), HRP-conjugated anti-FLAG M2 (Sigma-Aldrich, #A8592, RRID:AB_439702), and HRP-conjugated anti-HA (Roche, #12013819001, RRID:AB_390917). Secondary antibodies used in this work include HRP-conjugated anti-rabbit IgG (Sigma-Aldrich, #A6154, RRID:AB_258284) and HRP-conjugated anti-mouse IgG (Sigma-Aldrich, #A6782, RRID:AB_258315). All experiments were run in biological triplicates. Intensities of protein bands on immunoblots were quantified using ImageJ.

## Mass spectrometry

Mass spectrometry was carried out as previously described (*Wang et al., 2023*). Immunoprecipitates on protein A/G beads were snap-frozen and stored at –70 °C. Peptides were pre-fractionated using high-pH fractionation and analyzed on the Thermo Ultimate 3000 RSLCnano System via direct injection. Data were processed using MaxQuant/Andromeda (version 1.6.2.10) and compared to Uniprot-annotated protein sequences. False discovery rates were set to 0.01 for protein and peptide assignment with a minimum peptide length of four residues and a minimum peptide number of one.

## Structural prediction and analysis

The structural model of the ARF1-GTP:COMMD3-NTD (a.a. 1–120) heterodimer was predicted using AlphaFold3 with default settings (*Abramson et al., 2024*). Five independent structural models were generated for each protein complex, and the quality of the predicted models was assessed through their interface predicted template modeling (ipTM) scores, predicted template modeling (pTM) scores, predicted alignment error (PAE) plots, and predicted local distance difference test (pLDDT) scores (*Abramson et al., 2024*; *Mirdita et al., 2022*). Structural analysis was conducted using UCSF ChimeraX (v1.8) (*Goddard et al., 2018*) and PDBePISA (*Krissinel and Henrick, 2007*).

## Statistical analysis

All data shown in the figures are from at least three independent biological replicates. Biological replicates were plated, treated, and analyzed in parallel. Statistical analyses were performed using GraphPad Prism 10. Student's $t$-tests were used for comparisons between two groups. One-way ANOVA with Dunnett's multiple comparisons test (comparing all groups to control) or Sidak's multiple comparisons test (for internal group comparisons) was used when analyzing experiments with more than two groups. Significance is indicated as follows: n.s., not significant; *$p<0.05$; **$p<0.01$; ***$p<0.001$. Additional details on sample size, error bars, and statistical tests are provided in the figure legends.

## Acknowledgements

We thank Drs. Santiago Di Pietro, Da Jia, Andrea Ambrosio, Greg Odorizzi, Gia Voeltz, and Mitchell Leih for reagents or helpful suggestions. We thank Yan Ouyang, Jingyi Wu, James Orth, and Christopher C Ebmeier for technical assistance. This work was supported by National Institutes of Health grants GM126960 (JS) and DK124431 (JS). Publication of this article was partially funded by the University of Colorado Boulder Libraries Open Access Fund.

## Additional information

### Funding

| Funder | Grant reference number | Author |
| --- | --- | --- |
| National Institutes of Health | GM126960 | Jingshi Shen |
| National Institutes of Health | DK124431 | Jingshi Shen |

| Funder | Grant reference number | Author |
|---|---|---|

The funders had no role in study design, data collection and interpretation, or the decision to submit the work for publication.

## Author contributions

Galen T Squiers, Conceptualization, Data curation, Formal analysis, Validation, Investigation, Visualization, Methodology, Writing - original draft, Writing – review and editing; Chun Wan, Harrison Puscher, Investigation, Writing – review and editing; James Gorder, Investigation, Methodology; Jingshi Shen, Conceptualization, Resources, Supervision, Funding acquisition, Validation, Writing - original draft, Project administration, Writing – review and editing

## Author ORCIDs

Jingshi Shen ⓘ https://orcid.org/0000-0001-9595-1148

Reviewer #1 (Public review): https://doi.org/10.7554/eLife.105264.3.sa1
Reviewer #2 (Public review): https://doi.org/10.7554/eLife.105264.3.sa2
Reviewer #3 (Public review): https://doi.org/10.7554/eLife.105264.3.sa3
Author response https://doi.org/10.7554/eLife.105264.3.sa4

# Additional files

## Supplementary files

MDAR checklist

Supplementary file 1. The interactome of COMMD3.

Source data 1. Original CIF file of the structural model shown in *Figure 8A*.

## Data availability

All data generated or analysed during this study are included in the manuscript and supporting files. The data generated or analyzed in this study are included in the manuscript and supporting files. Source data files for *Figures 1–8*, *Figure 4—figure supplement 1*, *Figure 5—figure supplement 1*, *Figure 6—figure supplement 1* are provided, containing numerical datasets and uncropped immunoblots. Materials generated in this study will be made available upon request under material transfer agreements.

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
