## [Editor Report · eLife Assessment]

This **important** study explores the mechanisms underlying the maintenance of cell surface protein levels. The authors present **solid** evidence to support their claims, though the addition of certain validation experiments could have further strengthened the conclusions. This work will be of particular interest to cell biologists focused on membrane trafficking.

---

## [Referee Report · Reviewer #1 (Public review)]

G. Squiers et al. analyzed a previously reported CRISPR genetic screening dataset of engineered GLUT4 cell-surface presentation and identified the Commander complex subunit COMMD3 as being required for endosomal recycling of specific cargo protein, transferrin receptor (TfR), to the cell surface. Through comparison of COMMD3-KO and other Commander subunit-KO cells, they demonstrated that the role of COMMD3 in mediating TfR recycling is independent of the Commander complex. Structural analysis and co-immunoprecipitation followed by mass spectrometry revealed that TfR recycling by COMMD3 relies on ARF1. COMMD3 interacts with ARF1 through its N-terminal domain (NTD) to stabilize ARF1. A mutation in the NTD of COMMD3 failed to rescue cell surface TfR in COMMD3-KO cells. In conclusion, the authors assert that COMMD3 stabilizes ARF1 in a Commander complex-independent manner, which is essential for recycling specific cargo proteins from endosomes to the plasma membrane.

The conclusions of this paper are generally supported by data, but some validation experiments should be included to strengthen the study.

(1) Specific role of ARF1 to COMMD3:

The authors don't think KO/KD of ARF1 is appropriate to address its specificity to COMMD3 cargo selection, so they focused on the COMMD3 NTD mutant. Though the mutant failed to rescue COMMD3 cargo TfR recycling, they did not examine the Commander cargo ITGA6. In addition, they cannot validate that the mutant interrupts the interaction between NTD and ARF1. These missing results and validation make their claim that ARF1 is specific to the COMMD3's Commander-independent function less convincing.

---

## [Referee Report · Reviewer #2 (Public review)]

Summary:

The Commander complex is a key player in endosomal recycling which recruits cargo proteins and facilitates the formation of tubulo-vesicular carriers. Squiers et al found COMMD3, a subunit of the Commander complex, could interact directly with ARF1 and regulate endosomal recycling.

Strengths:

Overall, this is a nice study that provides some interesting knowledge on the function of the Commander complex.

Comments on revisions:

The authors have addressed all my previous concerns

---

## [Referee Report · Reviewer #3 (Public review)]

Summary:

The study by Squiers and colleagues reveals a novel, Commander-independent role for COMMD3 in endosomal recycling. Through unbiased genetic screens, the authors identified COMMD3 as a regulator of GLUT4-SPR trafficking and validated its function using knockout experiments, which demonstrated its impact on endosomal morphology and trafficking independent of the Commander complex. Importantly, they mapped the interaction between the N-terminal domain (NTD) of COMMD3 and the GTPase Arf1, and through structure-guided mutagenesis, established that this interaction is essential for COMMD3's Commander-independent activity. The manuscript provides compelling evidence supporting this newly identified function of COMMD3, and I find the authors' interpretations well-justified. This is an excellent and intriguing study.

Comments on revisions:

The authors addressed all comments. Congratulations on this exciting work.

---

## [Author Response]

The following is the authors’ response to the original reviews

**Reviewer 1 (Public reviews):**
(1) Commander-Independent Role of COMMD3: While the authors provided evidence to support the Commander-independent role of COMMD3-such as the absence of other Commander subunits in the CRISPR screen and not decreased COMMD3 levels in other subunit-KO cells- direct evidence is lacking. The mutation that specifically disrupts the COMMD3-ARF1 interaction could serve as a valuable tool to directly address this question.

The Reviewer raised an excellent point. We fully agree with the Reviewer that multiple lines of evidence are needed to support the novel Commander-independent function of COMMD3.

Comparative genetic analyses in Figures 4 and 5 indicate that COMMD3 regulates endosomal retrieval independently of the Commander complex. In Figure 8 of the revised manuscript, we show that point mutations introduced into the COMMD3:ARF1 interface impair this Commander-independent function. Moreover, Figure 6 demonstrates that ARF1 upregulation fully rescues the KO phenotype of *COMMD3*. In addition, Figure S2 further supports that COMMD3 levels, but not those of other Commander subunits, correspond to its Commander-independent function in endosomal trafficking. We have also revised the Discussion section to elaborate on the implications of these findings. We appreciate the Reviewer’s advice.

(2) Role of ARF1 in Cargo Selection: The Commander-independent function of COMMD3 appears cargo-dependent and relies on ARF1's role in cargo selection. The authors should investigate whether KO/KD of ARF1 reduces cell surface levels of ITGA6 and TfR.

The Reviewer correctly pointed out that KO/KD of ARF1 may provide further insights into the Commander-independent function of COMMD3. However, since ARF1 is involved in cargo sorting at both the endosome and the *trans*-Golgi network, its KO would disrupt multiple trafficking routes, making the data difficult to interpret. Instead, we focused on point mutations in the NTD that specifically disrupt ARF1 binding without affecting the function of the Commander complex (Fig. 8). As these mutations impair the Commander-independent function of COMMD3, our data strongly support a direct role for ARF1 in this recycling pathway. We note that the discovery of a novel trafficking pathway inevitably opens many research directions. One such direction is to systematically identify cargoes that rely on COMMD3 but not the Commander complex for endosomal retrieval.

(3) Impact on TfR Stability: Figure 7D suggests that TfR protein levels are reduced in COMMD3-KO cells, potentially due to degradation caused by disrupted recycling. This raises the question of whether the observed reduction in cell surface TfR is due to impaired endosomal recycling or decreased total protein levels. The authors should quantify the ratio of cell surface protein to total protein for TfR, GLUT-SPR, and ITGA6 in COMMD3-KO cells.

Based on the Reviewer's suggestion, we quantified both the total levels and the surface-tototal ratio of TfR, as shown in Figure S1 of the revised manuscript. These new data further support the conclusion that defects in TfR retrieval lead to its lysosomal degradation. The GLUT-SPR data presented in the main figures represent the surface-to-total ratio of the GLUT-SPR reporter. We thank the Reviewer for the important suggestion.

**Reviewer #1 (Recommendations for the authors):**
(1) Commander-Independent Role of COMMD3: The mutation that specifically disrupts the COMMD3-ARF1 interaction could serve as a valuable tool to directly address this question. The authors should evaluate whether the full-length mutant of COMMD3 can rescue decreased levels of CCDC93 and VPS35L, as well as cell surface ITGA6, TfR, and GLUT4 inCOMMD3-KO cells.

This is an excellent point. In our mechanistic experiments, we focused on the NTD of COMMD3 because this domain mediates its Commander-independent function and is not involved in forming the Commander holo-complex. This approach allowed us to draw unambiguous conclusions. Nevertheless, we anticipate that full-length COMMD3 carrying these point mutations would also be defective in regulating Commander-independent cargo.

(2) Role of ARF1 in Cargo Selection: The authors should investigate whether KO/KD of ARF1 reduces cell surface levels of ITGA6 and TfR. Was ARF1 identified in the initial CRISPR screen? If so, this should be explicitly noted. Alternatively, does ARF1 overexpression rescue ITGA6 levels in COMMD3-KO cells? Furthermore, does ARF1 overexpression rescue TfR levels in COMMD3 and CCDC93 double-KO cells?

Reinto the Commander-independent function of COMMD3. However, since ARF1 is involved in cargo sorting at both the endosome and the *trans*-Golgi network, its KO would disrupt multiple trafficking routes, making the data difficult to interpret. Instead, we focused on point mutations that specifically disrupt ARF1 binding without affecting the function of the Commander complex (Fig. 8). Since these mutations impair the Commander-independent function of COMMD3, our data strongly support a direct role for ARF1 in this novel recycling pathway. Based on our genetic data, we anticipate that all COMMD3-dependent cargoes will be similarly rescued in ARF1-overexpressing cells. In line with the Reviewer's comment, a key research direction we are currently pursuing is systematically determining how surface protein levels are affected by *COMMD3* KO and ARF1 overexpression using surface proteomics.

(3) Inconsistency in COMMD3 Rescue Levels (Figure 5A): Figure 5A shows comparable or higher levels of COMMD3 in rescued cells than in CCDC93-KO and VPS35L-KO cells. However, COMMD3 rescue did not increase cell surface TfR as much as in CCDC93-KO and VPS35L-KO cells. This inconsistency should be discussed or validated.

To address the Reviewer’s inquiry, we quantified COMMD3 expression levels in these cell lines using multiple independent experiments. The new data are presented in Figure S2 of the revised manuscript. These expanded datasets allowed us to more accurately determine the relationship between COMMD3 expression and our genetic data. Since the Commander complex remains intact in the COMMD3 rescue cells, a significant portion of COMMD3 proteins are expected to be incorporated into the Commander complex, which does not regulate TfR recycling. In contrast, because the Commander complex is disrupted in *Ccdc93* and *Vps35l* KO cells, all COMMD3 proteins are available to regulate TfR recycling in a Commander-independent manner. These findings are fully consistent with the similar surface TfR levels observed in *Ccdc93/Vps35l* KO cells and *COMMD3* overexpressing cells. We thank the Reviewer for this excellent suggestion.

(4) Significance of NTD in COMMD3 Function: The conclusion that "the NTD of COMMD3 mediates its Commander-independent function and interacts with ARF1" (Page 12) is not fully supported without a side-by-side comparison of NTD, CTD, and FL COMMD3 in the same experiment (e.g., Figures 6B and 6G). Additional data is needed to strengthen this claim.

We conducted the experiment suggested by the Reviewer and included the data in Figure S3. Our results indicate that the COMMD3 CTD cannot mediate the Commander-independent function of COMMD3 in endosomal retrieval. We appreciate the Reviewer’s suggestion.

(5) ARF1 Stabilization Experiments: To substantiate the claim that COMMD3 binds and stabilizes the GTP-form of ARF1, the authors should include a comparative experiment showing GTP-form, GDPform, and wild-type ARF1 (e.g., Figures 6G and 7C).

We fully agree with the Reviewer that it would be important to compare how the ARF1:COMMD3 interaction is influenced by the nucleotide-binding state. However, trapping ARF1 in its GDP-bound state remains unfeasible, and nucleotide-free small GTPases are inherently unstable. In addition, WT ARF1 likely exists as a mixture of GTP- and GDP-bound forms, further complicating the analysis. To address the Reviewer’s comment, we used AlphaFold3 predictions. Interestingly, we found that the ipTM score of GTP-ARF1:COMMD3 is significantly higher than that of GDP-ARF1:COMMD3 or apo-ARF1:COMMD3, supporting our conclusion that COMMD3 recognizes and stabilizes the active form of ARF1.

(6) Validation of NTD Mutation (Figure 8): Co-immunoprecipitation or cellular co-localization experiments should be performed to confirm that the NTD mutation disrupts the interaction between COMMD3 and ARF1, as depicted in Figure 8.

This is an important question, and the best approach to address it would be to measure the binding affinity of the WT and mutant proteins using ITC or SPR. However, this is currently unfeasible, as we have not yet obtained pure recombinant COMMD3 and GTP-ARF1 proteins. Co-IP, by nature, is a crude assay that often fails to detect changes in binding affinity. A previous study on other proteins showed that mutations in protein-binding interfaces strongly reduced binding affinity as measured by SPR, but these changes would have been missed by co-IP assays (PMID: 25500532). In agreement with this limitation, our co-IP experiments did not yield conclusive results. Instead, we focused on structure-guided genetic experiments, which unequivocally demonstrated the effects of targeted mutations on the Commander-independent function of COMMD3.

**Reviewer #2 (Public review):**
(1) All existing data suggest that COMMD3 is a subunit of the Commander complex. Is there any evidence that COMMD3 can exist as a monomer?

The Reviewer raised an intriguing point. Indeed, COMMD proteins, including COMMD3, can exist outside the Commander holo-complex and form homo- or hetero-oligomers, as monomeric COMMD proteins are likely unstable. These observations align well with the Commander-independent function identified in this study. We have revised the Discussion section of the manuscript to further elaborate on this point and thank the Reviewer for the suggestion.

(2) In Figure 9, the author emphasizes COMMD3-dependent cargo and Commander-dependent cargo. Can the authors speculate what distinguishes these two types of cargo? Do they contain sequence-specific motifs?

This is another important question. Our data clearly demonstrate that COMMD3 has a Commander-independent function in addition to its canonical role within the Commander holocomplex. Since cargo proteins typically possess multiple sorting signals that operate at different stages of the exocytic and endocytic pathways, identifying COMMD3-dependent sorting signals remains a challenge. ARF4 has been shown to specifically recognize the VXPX motif (PMID: 15728366), suggesting that ARF1 may similarly bind cytosolic sorting signals, with COMMD3 stabilizing this interaction. A key future direction is to systematically identify COMMD3-dependent cargo proteins and elucidate the mechanisms underlying their endosomal sorting. We have revised the Discussion section of the manuscript to explicitly address this point and thank the Reviewer for this important suggestion.

(3) What could be the possible mechanism underlying the observation that the knockout of COMMD3 results in larger early endosomes? How is the disruption of cargo retrieval related to the increase in endosome size?

The endosomal retrieval process is critical for recycling membrane proteins and lipids back to the plasma membrane or the *trans*-Golgi network. When this process is disrupted, cargo that should be recycled accumulates within endosomes, leading to their enlargement. For example, defects in retromer function can cause endosomal swelling due to cargo accumulation (PMID: 33380435). We added this citation to the revised manuscript and thank the Reviewer for the advice.

**Reviewer 3 (Recommendations for the authors):**
(1) Figure 4: How do the authors define Commander-dependent vs. Commander-independent cargos?In Figure 4, the surface expression of ITGA6 is reduced to approximately 0.75 across all knockouts. However, there is a similar level of reduction for GLUT4-SPR in the commd5 knockout and for LAMP1 in the commd5 and commd1 knockouts. Are GLUT4-SPR and LAMP1 Commander-dependent or Commander-independent cargos? Additionally, how does COMMD3 specifically identify/distinguish these cargos?

This is an excellent point. Our data suggest that TfR is a COMMD3-dependent but Commander-independent cargo, whereas ITGA6 is a Commander-dependent cargo that does not involve COMMD3-specific functions. The other two cargoes we examined—GLUT-SPR and LAMP1—primarily rely on COMMD3, with the Commander complex playing a minor role. Together, these observations clearly demonstrate that COMMD3 has a Commander-independent function in addition to its canonical role within the Commander holo-complex. Since cargo proteins typically possess multiple sorting signals that operate at different stages of the exocytic and endocytic pathways, identifying COMMD3-dependent sorting signals remains a challenge. ARF4 has been shown to specifically recognize the VXPX motif (PMID: 15728366), suggesting that ARF1 may similarly bind cytosolic sorting signals, with COMMD3 stabilizing this interaction. A key future direction is to systematically identify COMMD3-dependent cargo proteins and elucidate the mechanisms underlying their endosomal sorting. We have revised the Discussion section of the manuscript to explicitly address this point. We thank the Reviewer for this important suggestion.

(2) There is an increase in the surface expression of GLUT4-SPR in the commd1 knockout. Is this increase significant? The figure suggests a significant increase, but the text states it remains unchanged. Clarification is needed.

We found that surface levels of GLUT-SPR were slightly increased in *Commd1* KO cells, in stark contrast to the strong reduction observed in *Commd3* KO cells (Fig. 4B). This finding is consistent with our conclusion that COMMD3 has a distinct role from other Commander subunits. We have revised the Results section to more clearly describe these data and thank the Reviewer for the advice.

(3) Figure 5A: To support the claim that COMMD3 is upregulated in the vps35l KO/Ccdc93 KO, the authors should quantify COMMD3 expression. Also, why is there a Vps35l band present in the Vps35l knockout cells?

Based on the Reviewer’s suggestion, we quantified the total levels of COMMD3 and included these new data in Figure S2. In this study, gene deletion was achieved through the simultaneous introduction of two independent gRNAs. Based on our previous experience, this strategy typically results in the complete loss of gene expression. We posit that the residual band observed in *Vps35l* KO cells originates from background signals, such as nonspecific staining by the antibody.

(4) Figure 7: It is intriguing that COMMD3 stabilizes Arf1-GTP and can compensate for COMMD3 in knockout cells. However, is this stabilization specific to TfR cargo only? The authors should test additional Commander-dependent and Commander-independent cargos to clarify this point.

Based on our genetic data, we anticipate that all COMMD3-dependent cargoes will be similarly rescued in ARF1-overexpressing cells. In line with the Reviewer's comment, an important direction we are pursuing is the use of surface proteomics to systematically determine how surface protein levels are affected by *COMMD3* KO and ARF1 overexpression.

(5) Is Arf1 interaction specific to COMMD3? The authors should investigate the effects of Arf1 knockout on COMMD3 expression and test its role in regulating Commander-dependent and Commander-independent cargos.

The Reviewer raised an excellent point. Since ARF1 is involved in cargo sorting at both the endosome and the *trans*-Golgi network, its KO would interfere with multiple trafficking routes and the data would be difficult to interpret. Thus, in this work, we focused on the function and mechanism of the COMMD3:ARF1 complex on the endosome. Based on the suggestion of the Reviewer, we used AlphaFold3 to predict ARF1 binding to COMMD proteins. Interestingly, the complex with the highest predicted ipTM score is COMMD3:ARF1, while other COMMD proteins have much lower predicted binding scores. These results are consistent with the results of our unbiased CRISPR screens and targeted gene KO, and further support the conclusion that the COMMD3:ARF1 binding is specific and physiologically important in endosomal trafficking.